# Pancreatic progenitor epigenome maps prioritize type 2 diabetes risk genes with roles in development

Ryan J Geusz[1,2,3,4†], Allen Wang[1,2,3†], Joshua Chiou[1,4†], Joseph J Lancman[5,6†], Nichole Wetton[1,2,3], Samy Kefalopoulou[1,2,3], Jinzhao Wang[1,2,3], Yunjiang Qiu[2], Jian Yan[2], Anthony Aylward[1], Bing Ren[2,7], P Duc Si Dong[5,6], Kyle J Gaulton[1*], Maike Sander[1,2,3*]

[1]Department of Pediatrics, Pediatric Diabetes Research Center, University of California, San Diego, San Diego, United States; [2]Department of Cellular & Molecular Medicine, University of California, San Diego, San Diego, United States; [3]Sanford Consortium for Regenerative Medicine, San Diego, United States; [4]Biomedical Graduate Studies Program, University of California, San Diego, San Diego, United States; [5]Human Genetics Program, Sanford Burnham Prebys Medical Discovery Institute, San Diego, United States; [6]Graduate School of Biomedical Sciences, Sanford Burnham Prebys Medical Discovery Institute, San Diego, United States; [7]Ludwig Institute for Cancer Research, San Diego, United States

*For correspondence:
kgaulton@health.ucsd.edu (KJG);
masander@ucsd.edu (MS)

†These authors contributed equally to this work

**Abstract** Genetic variants associated with type 2 diabetes (T2D) risk affect gene regulation in metabolically relevant tissues, such as pancreatic islets. Here, we investigated contributions of regulatory programs active during pancreatic development to T2D risk. Generation of chromatin maps from developmental precursors throughout pancreatic differentiation of human embryonic stem cells (hESCs) identifies enrichment of T2D variants in pancreatic progenitor-specific stretch enhancers that are not active in islets. Genes associated with progenitor-specific stretch enhancers are predicted to regulate developmental processes, most notably tissue morphogenesis. Through gene editing in hESCs, we demonstrate that progenitor-specific enhancers harboring T2D-associated variants regulate cell polarity genes *LAMA1* and *CRB2*. Knockdown of *lama1* or *crb2* in zebrafish embryos causes a defect in pancreas morphogenesis and impairs islet cell development. Together, our findings reveal that a subset of T2D risk variants specifically affects pancreatic developmental programs, suggesting that dysregulation of developmental processes can predispose to T2D.

## Introduction

Type 2 diabetes (T2D) is a multifactorial metabolic disorder characterized by insulin insensitivity and insufficient insulin secretion by pancreatic beta cells (*Halban et al., 2014*). Genetic association studies have identified hundreds of loci influencing risk of T2D (*Mahajan et al., 2018*). However, disease-relevant target genes of T2D risk variants, the mechanisms by which these genes cause disease, and the tissues in which the genes mediate their effects remain poorly understood.

The majority of T2D risk variants map to non-coding sequence, suggesting that genetic risk of T2D is largely mediated through variants affecting transcriptional regulatory activity. Intersection of T2D risk variants with epigenomic data has uncovered enrichment of T2D risk variants in regulatory sites active in specific cell types, predominantly in pancreatic beta cells, including risk variants that affect regulatory activity directly (*Chiou et al., 2019*; *Fuchsberger et al., 2016*; *Gaulton et al.,*

*2015*; *Gaulton et al., 2010*; *Greenwald et al., 2019*; *Mahajan et al., 2018*; *Parker et al., 2013*; *Pasquali et al., 2014*; *Thurner et al., 2018*; *Varshney et al., 2017*). T2D risk-associated variants are further enriched within large, contiguous regions of islet active chromatin, referred to as stretch or super-enhancers (*Parker et al., 2013*). These regions of active chromatin preferentially bind islet-cell-restricted transcription factors and drive islet-specific gene expression (*Parker et al., 2013*; *Pasquali et al., 2014*).

Many genes associated with T2D risk in islets are not uniquely expressed in differentiated islet endocrine cells, but also in pancreatic progenitor cells during embryonic development. For example, T2D risk variants map to *HNF1A, HNF1B, HNF4A, MNX1, NEUROG3, PAX4,* and *PDX1* (*Flannick et al., 2019*; *Mahajan et al., 2018*; *Steinthorsdottir et al., 2014*), which are all transcription factors also expressed in pancreatic developmental precursors. Studies in model organisms and hESC-based models of pancreatic endocrine cell differentiation have shown that inactivation of these transcription factors causes defects in endocrine cell development, resulting in reduced beta cell numbers (*Gaertner et al., 2019*). Furthermore, heterozygous mutations for *HNF1A, HNF1B, HNF4A, PAX4,* and *PDX1* are associated with maturity onset diabetes of the young (MODY), which is an autosomal dominant form of diabetes with features similar to T2D (*Urakami, 2019*). Thus, there is evidence that reduced activity of developmentally expressed transcription factors can cause diabetes later in life.

The role of these transcription factors in T2D and MODY could be explained by their functions in regulating gene expression in mature islet cells. However, it is also possible that their function during endocrine cell development could predispose to diabetes instead of, or in addition to, endocrine cell gene regulation. One conceivable mechanism is that individuals with reduced activity of these transcription factors are born with either fewer beta cells or beta cells more prone to fail under conditions of increased insulin demand. Observations showing that disturbed intrauterine metabolic conditions, such as maternal malnutrition, can lead to reduced beta cell mass and T2D predisposition in the offspring (*Lumey et al., 2015*; *Nielsen et al., 2014*; *Portha et al., 2011*) support the concept that compromised beta cell development could predispose to T2D. However, whether there is T2D genetic risk relevant to the regulation of endocrine cell development independent of gene regulation in mature islet cells has not been explored.

In this study, we investigated the contribution of gene regulatory programs specifically active during pancreatic development to T2D risk. First, we employed a hESC-based differentiation system to generate chromatin maps of hESCs during their stepwise differentiation into pancreatic progenitor cells. We then identified T2D-associated variants localized in active enhancers in developmental precursors but not in mature islets, used genome editing in hESCs to define target genes of pancreatic progenitor-specific enhancers harboring T2D variants, and employed zebrafish genetic models to study the role of two target genes in pancreatic and endocrine cell development.

## Results

### Pancreatic progenitor stretch enhancers are enriched for T2D risk variants

To determine whether there is a development-specific genetic contribution to T2D risk, we generated genome-wide chromatin maps of hESCs during their stepwise differentiation into pancreatic progenitors through four distinct developmental stages: definitive endoderm (DE), gut tube (GT), early pancreatic progenitors (PP1), and late pancreatic progenitors (PP2) (*Figure 1A*). We then used ChromHMM (*Ernst and Kellis, 2012*) to annotate chromatin states, such as active promoters and enhancers, at all stages of hESC differentiation as well as in primary islets (*Figure 1—figure supplement 1A,B*).

Large and contiguous regions of active enhancer chromatin, which have been termed stretch- or super-enhancers (*Parker et al., 2013*; *Whyte et al., 2013*), are highly enriched for T2D risk variants in islets (*Parker et al., 2013*; *Pasquali et al., 2014*). We therefore partitioned active enhancers from each hESC developmental stage and islets into stretch enhancers (SE) and traditional (non-stretch) enhancers (TE) (*Figure 1B*). Consistent with prior observations of SE features (*Parker et al., 2013*; *Whyte et al., 2013*), SE comprised a small subset of all active enhancers (7.7%, 7.8%, 8.8%, 8.1%, 8.1%, and 10.4% of active enhancers in ES, DE, GT, PP1, PP2, and islets, respectively; *Figure 1B* and

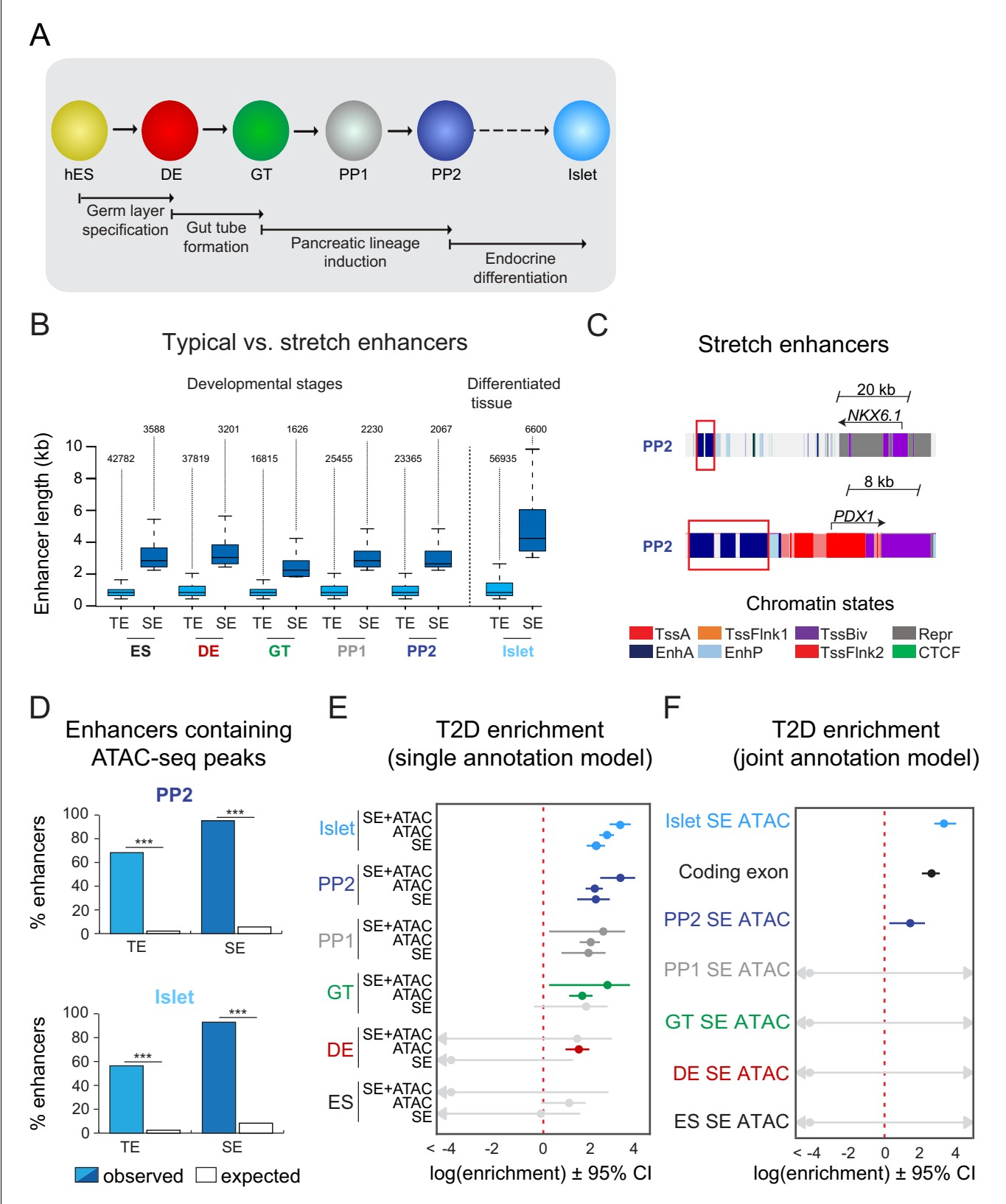

**Figure 1.** T2D-associated risk variants are enriched in stretch enhancers of pancreatic progenitors independent of islet stretch enhancers. (**A**) Schematic illustrating the stepwise differentiation of human embryonic stem cells (hES) into pancreatic progenitors (solid arrows) and lineage relationship to islets

*Figure 1 continued on next page*

*Figure 1 continued*

(dotted arrow). Developmental intermediates include definitive endoderm (DE), gut tube (GT), early pancreatic progenitor (PP1), and late pancreatic progenitor (PP2) cells. (B) Box plots depicting length of typical enhancers (TE) and stretch enhancers (SE) at each developmental stage and in primary human islets. Plots are centered on median, with box encompassing 25–75th percentile and whiskers extending up to 1.5 interquartile range. Total numbers of enhancers are shown above each box plot. (C) Examples of stretch enhancers (denoted with red boxes) near the genes encoding the pancreatic lineage-determining transcription factors NKX6.1 and PDX1, respectively. Chromatin states are based on ChromHMM classifications: TssA, active promoter; TssFlnk, flanking transcription start site; TssBiv, bivalent promoter; Repr, repressed; EnhA, active enhancer; EnhP, poised enhancer. (D) Percentage of TE and SE overlapping with at least one ATAC-seq peak at PP2 or in islets. Enrichment analysis comparing observed and expected overlap based on random genomic regions of the same size and located on the same chromosome averaged over 10,000 iterations (***$p<1 \times 10^{-4}$; permutation test). ATAC-seq peaks were merged from two independent differentiations for PP2 stage cells and four donors for primary islets. (E) Genome-wide enrichment of T2D-associated variants (minor allele frequency >0.0025) in stretch enhancers, ATAC-seq peaks, and ATAC-seq peaks within stretch enhancers for all developmental stages when modeling each annotation separately. Points and lines represent log-scaled enrichment estimates and 95% confidence intervals from functional genome wide association analysis (fgwas), respectively. ATAC-seq peaks were merged from two independent differentiations for ES, DE, GT, PP1, and PP2 stage cells and from four donors for primary islets. (F) Genome-wide enrichment of T2D-associated variants (minor allele frequency >0.0025) in ATAC-seq peaks within stretch enhancers for all developmental stages and coding exons when considering all annotations in a joint model. Points and lines represent log-scaled enrichment estimates and 95% confidence intervals from fgwas, respectively. ATAC-seq peaks were merged from two independent differentiations for ES, DE, GT, PP1, and PP2 stage cells and from four donors for primary islets. See also *Figure 1—figure supplement 1*.

The online version of this article includes the following figure supplement(s) for figure 1:

**Figure supplement 1.** Characterization of typical and stretch enhancers in pancreatic developmental intermediates and islets.

*Figure 1—figure supplement 1C*) and genes proximal to SE were more highly expressed than genes proximal to TE (p=$4.68 \times 10^{-7}$, $4.64 \times 10^{-11}$, $1.31 \times 10^{-5}$, $8.85 \times 10^{-9}$, $5.34 \times 10^{-6}$, and $<2.2 \times 10^{-16}$ for expression of genes near TE vs SE in ES, DE, GT, PP1, PP2, and islets, respectively; *Figure 1—figure supplement 1D*). Genes near SE in pancreatic progenitors included transcription factors involved in the regulation of pancreatic cell identity, such as *NKX6.1* and *PDX1* (*Figure 1C*). Since disease-associated variants are preferentially enriched in narrow peaks of accessible chromatin within broader regions of active chromatin (*Greenwald et al., 2019*; *Thurner et al., 2018*; *Varshney et al., 2017*), we next used ATAC-seq to generate genome-wide maps of chromatin accessibility across all time points of differentiation. Nearly all identified SE contained at least one ATAC-seq peak (*Figure 1D* and *Figure 1—figure supplement 1E,F*). At the PP2 stage, 62.3% of SE harbored one, 32.2% two or three, and 0.7% four or more ATAC-seq peaks (*Figure 1—figure supplement 1F*). Similar percentages were observed in earlier developmental precursors and islets.

Having annotated accessible chromatin sites within SE, we next tested for enrichment of T2D-associated variants in SE active in mature islets and in pancreatic developmental stages. We observed strongest enrichment of T2D-associated variants in islet SE (log enrichment = 2.18, 95% CI = 1.80, 2.54) and late pancreatic progenitor SE (log enrichment = 2.17, 95% CI = 1.40, 2.74), which was more pronounced when only considering variants in accessible chromatin sites within these elements (islet log enrichment = 3.20, 95% CI = 2.74, 3.60; PP2 log enrichment = 3.18, 95% CI = 2.35, 3.79; *Figure 1E*). Given that a subset of pancreatic progenitor SE is also active in islets, we next determined whether pancreatic progenitor SE contribute to T2D risk independently of islet SE. Variants in accessible chromatin sites of late pancreatic progenitor SE were enriched for T2D association in a joint model including islet SE (islet log enrichment = 2.94, 95% CI = 2.47, 3.35; PP2 log enrichment = 1.27, 95% CI = 0.24, 2.00; *Figure 1F*). We also observed enrichment of variants in accessible chromatin sites of pancreatic progenitor SE after conditioning on islet SE (log enrichment = 0.60, 95% CI = −0.87, 1.48), as well as when excluding pancreatic progenitor SE active in islets (log enrichment = 1.62, 95% CI = <-20, 3.14). Examples of known T2D loci with T2D-associated variants in SE active in pancreatic progenitors but not in islets included *LAMA1* and *PROX1*. These results suggest that a subset of T2D variants may affect disease risk by altering regulatory programs specifically active in pancreatic progenitors.

## Pancreatic progenitor-specific stretch enhancers are near genes that regulate tissue morphogenesis

Having observed enrichment of T2D risk variants in pancreatic progenitor SE independent of islet SE, we next sought to further characterize the regulatory programs of SE with specific function in

pancreatic progenitors. We therefore defined a set of pancreatic progenitor-specific stretch enhancers (PSSE) based on the following criteria: (i) annotation as a SE at the PP2 stage, (ii) no classification as a SE at the ES, DE, and GT stages, and (iii) no classification as a TE or SE in islets. Applying these criteria, we identified a total of 492 PSSE genome-wide (*Figure 2A* and *Figure 2—source data 1*).

As expected based on their chromatin state classification, PSSE acquired broad deposition of the active enhancer mark H3K27ac at the PP1 and PP2 stages (*Figure 2B,C*). Coincident with an increase in H3K27ac signal, chromatin accessibility at PSSE also increased (*Figure 2B*), and 93.5% of PSSE contained at least one accessible chromatin site at the PP2 stage (*Figure 2—figure supplement 1A, B*). Further investigation of PSSE chromatin state dynamics at earlier stages of pancreatic differentiation revealed that PSSE were often poised (defined by H3K4me1 in the absence of H3K27ac) prior to activation (42%, 48%, 63%, and 17% of PSSE in ES, DE, GT, and PP1, respectively; *Figure 2C*), consistent with earlier observations that a poised enhancer state frequently precedes enhancer activation during development (*Rada-Iglesias et al., 2011*; *Wang et al., 2015*). Intriguingly, a subset of PSSE was classified as TE earlier in development (13%, 23%, 29%, and 46% of PSSE in ES, DE, GT, and PP1, respectively; *Figure 2C*), suggesting that SE emerge from smaller regions of active chromatin seeded at prior stages of development. During differentiation into mature islet cells, PSSE lost H3K27ac but largely retained H3K4me1 signal (62% of PSSE) (*Figure 2C*), persisting in a poised state in terminally differentiated islet cells.

To gain insight into the transcription factors that regulate PSSE, we conducted motif enrichment analysis of accessible chromatin sites within PSSE (*Figure 2—figure supplement 1C*). Consistent with the activation of PSSE upon pancreas induction, motifs associated with transcription factors known to regulate pancreatic development (*Conrad et al., 2014*; *Masui et al., 2007*) were enriched, including FOXA (p=1 $\times$ 10$^{-34}$), PDX1 (p=1 $\times$ 10$^{-30}$), GATA (p=1 $\times$ 10$^{-25}$), ONECUT (p=1 $\times$ 10$^{-17}$), and RBPJ (p=1 $\times$ 10$^{-14}$), suggesting that pancreatic lineage-determining transcription factors activate PSSE. Analysis of the extent of PSSE overlap with ChIP-seq binding sites for FOXA1, FOXA2, GATA4, GATA6, PDX1, HNF6, and SOX9 at the PP2 stage substantiated this prediction (p<1 $\times$ 10$^{-4}$ for all transcription factors; permutation test; *Figure 2D*).

Annotation of biological functions of predicted target genes for PSSE (nearest gene with FPKM $\geq$1 at PP2 stage) revealed gene ontology terms related to developmental processes, such as tissue morphogenesis (p=1 $\times$ 10$^{-7}$) and vascular development (p=1 $\times$ 10$^{-8}$), as well as developmental signaling pathways, including BMP (p=1 $\times$ 10$^{-5}$), NOTCH (p=1 $\times$ 10$^{-4}$), and canonical Wnt signaling (p=1 $\times$ 10$^{-4}$; *Figure 2E* and *Figure 2—source data 2*), which have demonstrated roles in pancreas morphogenesis and cell lineage allocation (*Ahnfelt-Rønne et al., 2010*; *Li et al., 2015*; *Murtaugh, 2008*; *Sharon et al., 2019*; *Sui et al., 2013*). Consistent with the temporal pattern of H3K27ac deposition at PSSE, transcript levels of PSSE-associated genes increased upon pancreatic lineage induction and peaked at the PP2 stage (p=1.8 $\times$ 10$^{-8}$; *Figure 2—figure supplement 1D*). Notably, expression of these genes sharply decreased in islets (p<2.2 $\times$ 10$^{-16}$), underscoring the likely role of these genes in regulating pancreatic development but not mature islet function.

## Pancreatic progenitor-specific stretch enhancers are highly specific across T2D-relevant tissues and cell types

We next sought to understand the phenotypic consequences of PSSE activity in the context of T2D pathophysiology. Variants in accessible chromatin sites of PSSE genome-wide were enriched for T2D association (log enrichment = 2.85, 95% CI = <-20, 4.09). We determined enrichment of genetic variants for T2D-related quantitative endophenotypes within accessible chromatin sites of PSSE, as well as all pancreatic progenitor SE (not just progenitor-specific) and islet SE, using LD score regression (*Bulik-Sullivan et al., 2015*; *Finucane et al., 2015*). As expected based on prior observations (*Parker et al., 2013*; *Pasquali et al., 2014*), we observed enrichment (Z > 1.96) of variants associated with quantitative traits related to insulin secretion and beta cell function within islet SE, exemplified by fasting proinsulin levels, HOMA-B, and acute insulin response (Z = 2.8, Z = 2.6, and Z = 2.2, respectively; *Figure 2F*). Conversely, PSSE showed a trend toward depletion for these traits, although the estimates were not significant. We further tested for enrichment in the proportion of variants in PSSE and islet SE nominally associated (p<0.05) with beta cell function traits compared to background variants. There was significant enrichment of beta cell trait association among islet SE

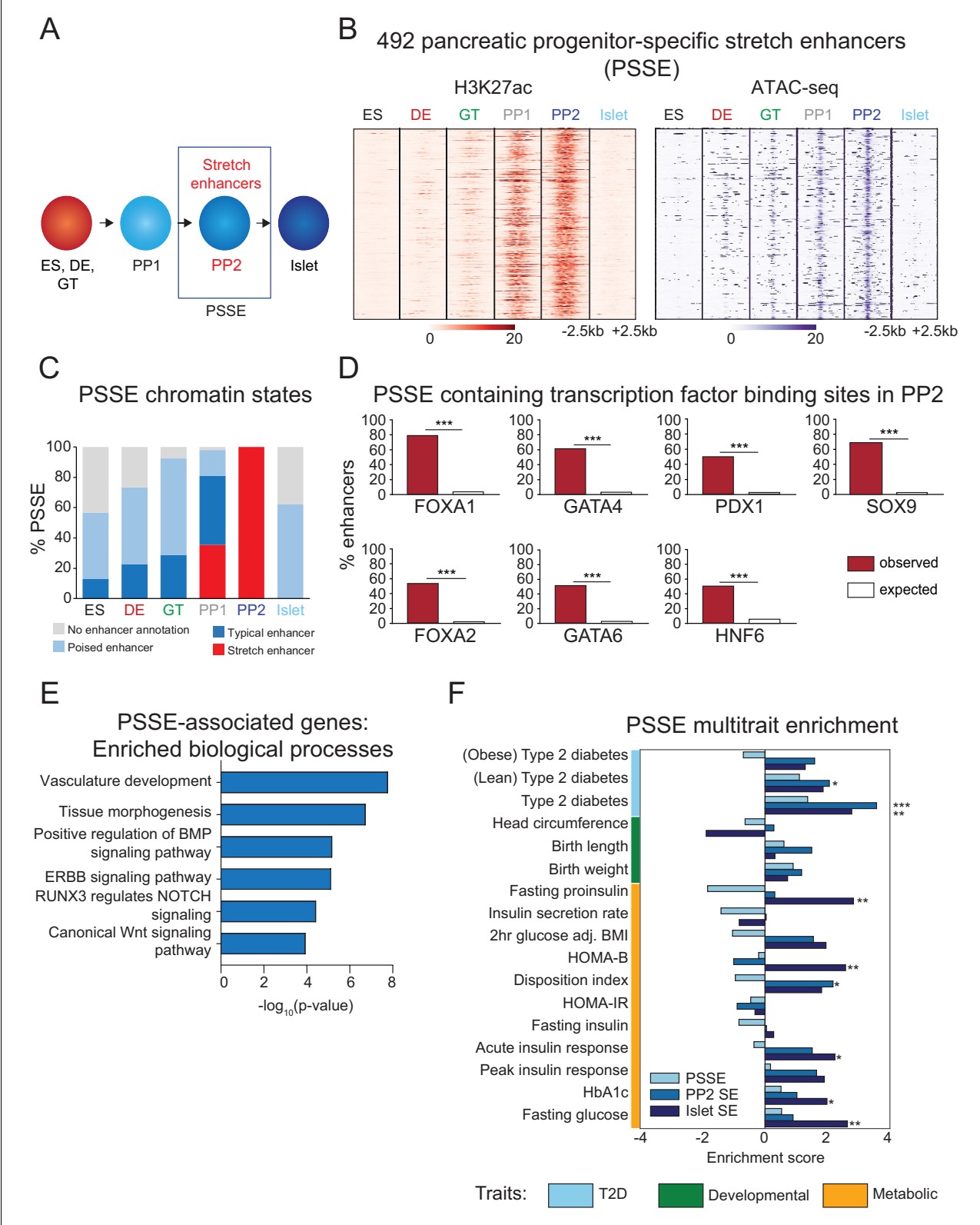

**Figure 2.** Candidate target genes of pancreatic progenitor-specific stretch enhancers regulate developmental processes. (**A**) Schematic illustrating identification of pancreatic progenitor-specific stretch enhancers (PSSE). (**B**) Heatmap showing density of H3K27ac ChIP-seq and ATAC-seq reads at PSSE, centered on overlapping H3K27ac and ATAC-seq peaks, respectively, and spanning 5 kb in ES, DE, GT, PP1, PP2, and islets. PSSE coordinates in *Figure 2—source data 1*. (**C**) Percentage of PSSE exhibiting indicated chromatin states at defined developmental stages and in islets. (**D**) Percentage

*Figure 2 continued on next page*

*Figure 2 continued*

of PSSE overlapping with at least one ChIP-seq peak at PP2 for the indicated transcription factors. Enrichment analysis comparing observed and expected overlap based on random genomic regions of the same size and located on the same chromosome averaged over 10,000 iterations (***p<1×10$^{-4}$; permutation test). (E) Gene ontology analysis for nearest expressed genes (fragments per kilobase per million fragments mapped (FPKM) ≥1 at PP2) to the 492 PSSE. See also *Figure 2—source data 2*. (F) Enrichment (LD score regression coefficient z-scores) of T2D, developmental, and metabolic GWAS trait-associated variants at accessible chromatin sites in PSSE as compared with PP2 and islet stretch enhancers. Significant enrichment was identified within accessible chromatin at PP2 stretch enhancers for lean type 2 diabetes (Z = 2.06, *p=3.94 × 10$^{-2}$), at PP2 stretch enhancers for type 2 diabetes (Z = 3.57, ***p=3.52 × 10$^{-4}$), at islet stretch enhancers for type 2 diabetes (Z = 2.78, **p=5.46 × 10$^{-3}$), at islet stretch enhancers for fasting proinsulin levels (Z = 2.83, **p=4.61 × 10$^{-3}$), at islet stretch enhancers for HOMA-B (Z = 2.58, **p=9.85 × 10$^{-3}$), at PP2 stretch enhancers for disposition index (Z = 2.18, *p=2.94 × 10$^{-2}$), at islet stretch enhancers for acute insulin response (Z = 2.24, *p=2.51 × 10$^{-2}$), at islet stretch enhancers for HbA1c (Z = 1.98, *p=4.72 × 10$^{-2}$), and at islet stretch enhancers for fasting glucose levels (Z = 2.64, **p=8.31 × 10$^{-3}$). See also *Figure 2—source data 3* and *Figure 2—figure supplement 1*.

The online version of this article includes the following source data and figure supplement(s) for figure 2:

**Source data 1.** Chromosomal coordinates of pancreatic progenitor-specific stretch enhancers (PSSE).
**Source data 2.** Enriched gene ontology terms for PSSE-associated genes.
**Source data 3.** Proportion of variants nominally associated with beta cell functional traits.
**Source data 4.** Tissue identity of downloaded data from ROADMAP consortium.
**Figure supplement 1.** Characterization of pancreatic progenitor-specific stretch enhancers.

variants ($\chi^2$ test; p<0.05 for all beta cell functional traits except for insulin secretion rate), but no corresponding enrichment for PSSE (*Figure 2—source data 3*).

A prior study found that variants at the *LAMA1* locus had stronger effects on T2D risk among lean relative to obese cases (*Perry et al., 2012*). Since we identified a PSSE at the *LAMA1* locus, we postulated that variants in PSSE collectively might have differing impact on T2D risk in cases segregated by BMI. We therefore tested PSSE, as well as pancreatic progenitor SE and islet SE, for enrichment of T2D association using GWAS of lean and obese T2D (*Perry et al., 2012*), using LD score regression (*Bulik-Sullivan et al., 2015*; *Finucane et al., 2015*). We observed nominally significant enrichment of variants in pancreatic progenitor SE for T2D among lean cases (Z = 2.1). Variants in PSSE were mildly enriched for T2D among lean (Z = 1.1) and depleted among obese (Z = −0.70) cases, although neither estimate was significant. By comparison, islet SE showed positive enrichment for T2D among both lean (Z = 1.9) and obese cases (Z = 1.3; *Figure 2F*). Together, these results suggest that PSSE may affect T2D risk in a manner distinct from islet SE function.

Having observed little evidence for enrichment of PSSE variants for traits related to beta cell function, we asked whether the enrichment of PSSE for T2D-associated variants could be explained by PSSE activity in T2D-relevant tissues and cell types outside the pancreas. We assessed PSSE activity by measuring H3K27ac signal in 95 representative tissues and cell lines from the ENCODE and Epigenome Roadmap projects (*Kundaje et al., 2015*). Interestingly, there was group-wide specificity of PSSE to pancreatic progenitors relative to other cells and tissues including those relevant to T2D, such as adipose tissue, skeletal muscle, and liver (*Figure 2—figure supplement 1E* and *Figure 2—source data 4*). Since gene regulation in adipocyte precursors also contributes to T2D risk (*Claussnitzer et al., 2014*), we further examined PSSE specificity with respect to chromatin states during adipogenesis, using data from human adipose stromal cell differentiation stages (hASC1-4) (*Mikkelsen et al., 2010*; *Varshney et al., 2017*). PSSE exhibited virtually no active chromatin during adipogenesis (9, 8, 6, and 8 out of the 492 PSSE were active enhancers in hACS-1, hASC-2, hASC-3, and hASC-4, respectively; *Figure 2—figure supplement 1F*). These findings identify PSSE as highly pancreatic progenitor-specific across T2D-relevant tissues and cell types.

## Identification of pancreatic progenitor-specific stretch enhancers harboring T2D-associated variants

Given the relative specificity of PSSE to pancreatic progenitors, we next sought to identify T2D-associated variants in PSSE at specific loci which may affect pancreatic development. We therefore identified variants in PSSE with evidence of T2D association (at p=4.7 × 10$^{-6}$) after correcting for the total number of variants in PSSE genome-wide (n = 10,738). In total there were 49 variants in PSSE with T2D association exceeding this threshold mapping to 11 loci (*Figure 3A*). This included variants at nine loci with known genome-wide significant T2D association (*PROX1*, *ST6GAL1*, *SMARCAD1*,

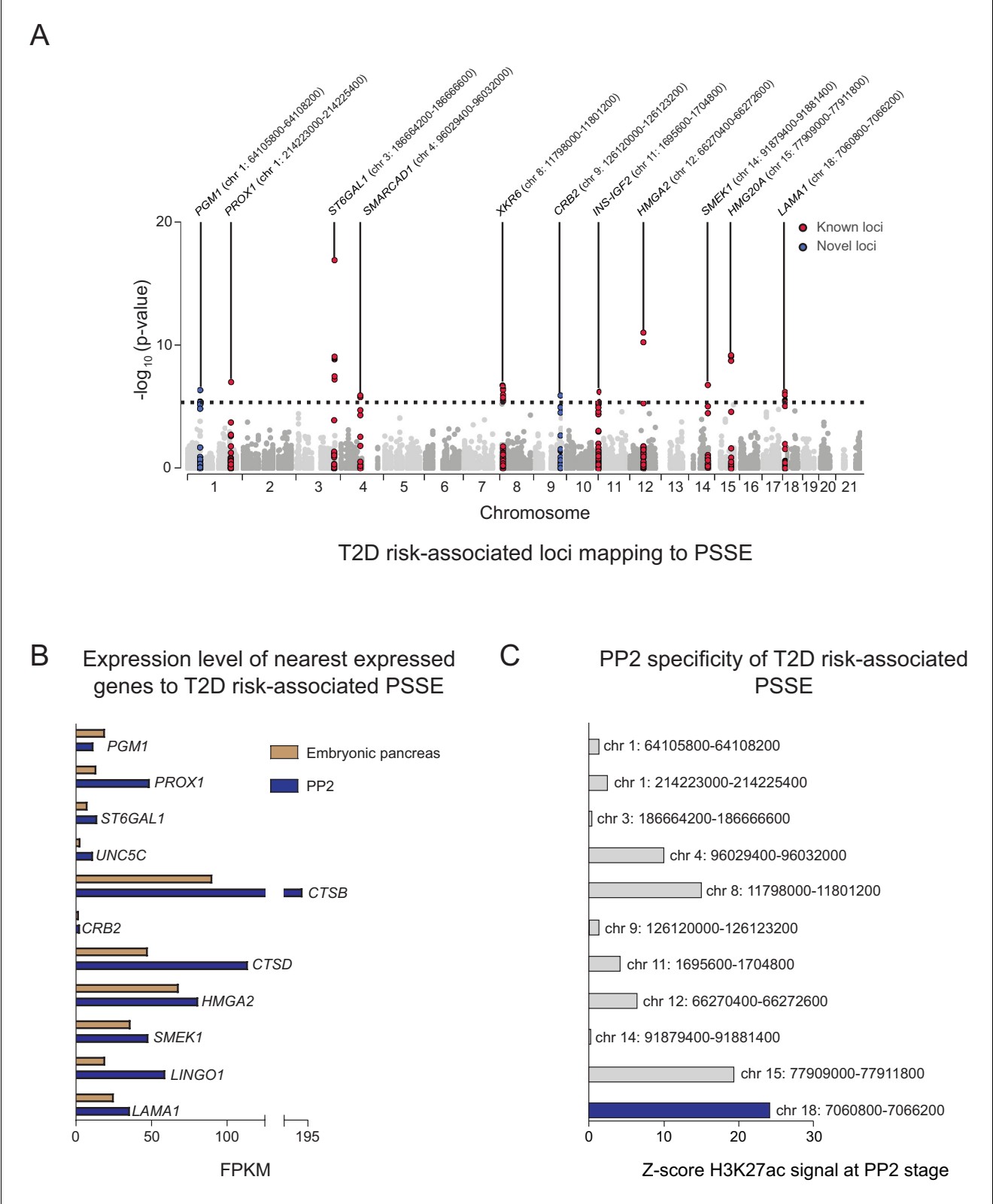

**Figure 3.** Identification of T2D risk variants associated with pancreatic progenitor-specific stretch enhancers. (**A**) Manhattan plot showing T2D association p-values (from **Mahajan et al., 2018**) for 10,738 variants mapping within PSSE. The dotted line shows the threshold for Bonferroni correction (p=4.66 × 10$^{-6}$). Novel loci identified with this threshold and mapping at least 500 kb away from a known locus are highlighted in blue. Chromosomal coordinates of T2D-associated PSSE are indicated. (**B**) mRNA levels (measured in fragments per kilobase per million fragments mapped

*Figure 3 continued on next page*

*Figure 3 continued*

[FPKM]) at PP2 (blue) and in human embryonic pancreas (54 and 58 days gestation, gold) of nearest expressed (FPMK ≥1) gene at PP2 for PSSE harboring T2D variants identified in A. (**C**) PP2 specificity of H3K27ac signal at PSSE harboring T2D variants identified in A. Z-score comparing H3K27ac signal at PP2 to H3K27ac signal in tissues and cell lines from the ENCODE and Epigenome Roadmap projects. See also ***Figure 3—figure supplement 1***.

The online version of this article includes the following figure supplement(s) for figure 3:

**Figure supplement 1.** Activity of T2D risk-associated pancreatic progenitor-specific stretch enhancers across human tissues.

*XKR6*, *INS-IGF2*, *HMGA2*, *SMEK1*, *HMG20A*, and *LAMA1*), as well as at two previously unreported loci with sub-genome-wide significant association, *CRB2* and *PGM1*. To identify candidate target genes of the T2D-associated PSSE in pancreatic progenitors, we analyzed the expression of all genes within the same topologically associated domain (TAD) as the PSSE in PP2 cells and in primary human embryonic pancreas tissue (*Figure 3B* and *Figure 3—figure supplement 1A*). These expressed genes are candidate effector transcripts of T2D-associated variants in pancreatic progenitors.

As many pancreatic progenitor SE remain poised in mature islets (*Figure 2C*), we considered whether T2D-associated variants in PSSE could have gene regulatory function in islets that is re-activated in the disease state. We therefore assessed overlap of PSSE variants with accessible chromatin of islets from T2D donors (*Khetan et al., 2018*). None of the strongly T2D-associated variants in PSSE (p=4.7 × 10$^{-6}$) overlapped an islet accessible chromatin site in T2D islets, arguing against the relevance of PSSE in broadly regulating islet gene activity during T2D.

## A pancreatic progenitor-specific stretch enhancer at *LAMA1* harbors T2D risk variants and regulates *LAMA1* expression selectively in pancreatic progenitors

Variants in a PSSE at the *LAMA1* locus were associated with T2D at genome-wide significance (*Figure 3A*), and *LAMA1* was highly expressed in the human embryonic pancreas (*Figure 3B*). Furthermore, the activity of the PSSE at the *LAMA1* locus was almost exclusively restricted to pancreatic progenitors (*Figure 3—figure supplement 1B,C*), and was further among the most progenitor-specific across all PSSE harboring T2D risk variants (*Figure 3C*). In addition, reporter gene assays in zebrafish embryos have shown that this enhancer drives gene expression specific to pancreatic progenitors in vivo (*Cebola et al., 2015*). We therefore postulated that the activity of T2D-associated variants within the *LAMA1* PSSE is relevant for gene regulation in pancreatic progenitors, and we sought to characterize the *LAMA1* PSSE in greater depth.

Multiple T2D-associated variants mapped within the *LAMA1* PSSE, and these variants were further in the 99% credible set in fine-mapping data from the DIAMANTE consortium (*Mahajan et al., 2018*; *Figure 4A*). No other variants in the 99% credible set mapped in an accessible chromatin site active in islets from either non-diabetic or T2D samples. The PSSE is intronic to the *LAMA1* gene and contains regions of poised chromatin and TE at prior developmental stages (*Figure 4A*). Consistent with its stepwise genesis as a SE throughout development, regions of open chromatin within the *LAMA1* PSSE were already present at the DE and GT stages. Furthermore, pancreatic lineage-determining transcription factors, such as FOXA1, FOXA2, GATA4, GATA6, HNF6, SOX9, and PDX1, were all bound to the PSSE at the PP2 stage (*Figure 4B*). Among credible set variants in the *LAMA1* PSSE, rs10502347 overlapped an ATAC-seq peak as well as ChIP-seq sites for multiple pancreatic lineage-determining transcription factors. Additionally, rs10502347 directly coincided with a SOX9 footprint identified in ATAC-seq data from PP2 cells, and the T2D risk allele C is predicted to disrupt SOX9 binding (*Figure 4B*). Consistent with the collective endophenotype association patterns of PSSE (*Figure 2F*), rs10502347 showed no association with beta cell function (p=0.81, 0.23, 0.46 for fasting proinsulin levels, HOMA-B, and acute insulin response, respectively; *Figure 4—figure supplement 1A*). Thus, T2D variant rs10502347 is predicted to affect the binding of pancreatic transcription factors and does not appear to affect beta cell function.

Enhancers can control gene expression over large genomic distances, and therefore their target genes cannot be predicted based on proximity alone. To directly assess the function of the *LAMA1* PSSE in regulating gene activity, we utilized CRIPSR-Cas9-mediated genome editing to generate

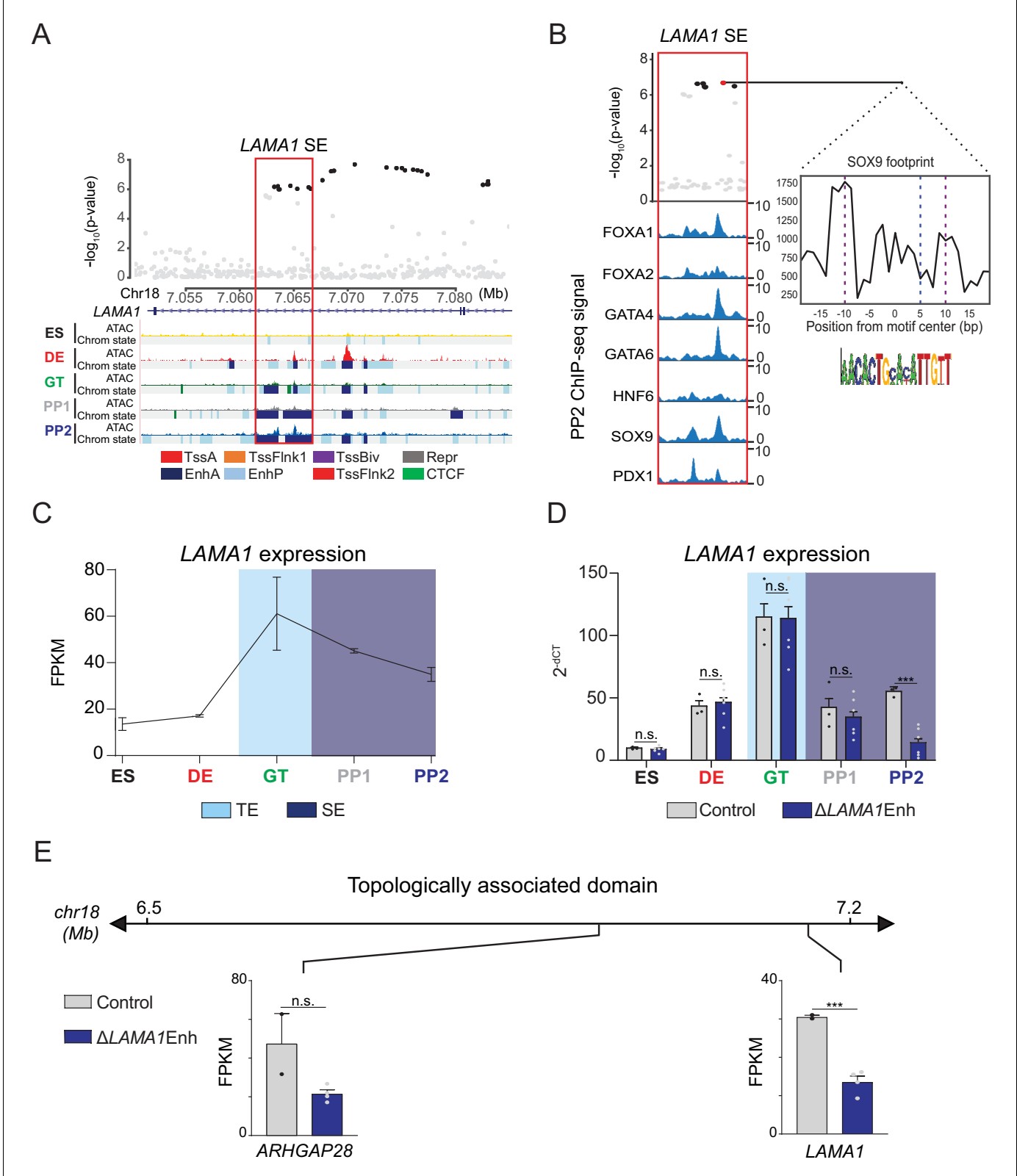

**Figure 4.** A T2D risk-associated *LAMA1* pancreatic progenitor-specific stretch enhancer regulates *LAMA1* expression specifically in pancreatic progenitors. (A) (Top) Locus plots showing T2D association p-values for variants in a 35 kb window (hg19 chr18:7,050,000–7,085,000) at the *LAMA1* locus and *LAMA1* PSSE (red box). Fine mapped variants within the 99% credible set for the *LAMA1* locus are colored black. All other variants are colored light gray. (Bottom) Chromatin states and ATAC-seq signal in ES, DE, GT, PP1, and PP2. TssA, active promoter; TssFlnk, flanking transcription start site;

*Figure 4 continued on next page*

Figure 4 continued

TssBiv, bivalent promoter; Repr, repressed; EnhA, active enhancer; EnhP, poised enhancer. (B) FOXA1, FOXA2, GATA4, GATA6, HNF6, SOX9, and PDX1 ChIP-seq profiles at the *LAMA1* PSSE in PP2. The variant rs10502347 (red) overlaps transcription factor binding sites and a predicted ATAC-seq footprint for the SOX9 sequence motif. Purple dotted lines indicate the core binding profile of the average SOX9 footprint genome-wide and the blue dotted line indicates the position of rs10502347 within the SOX9 motif. (C) *LAMA1* mRNA expression at each developmental stage determined by RNA-seq, measured in fragments per kilobase per million fragments mapped (FPKM). Data shown as mean ± S.E.M. (n = 3 replicates from independent differentiations). Light blue and purple indicate classification of the *LAMA1* PSSE as typical enhancer (TE) and stretch enhancer (SE), respectively. (D) *LAMA1* mRNA expression at each developmental stage determined by qPCR in control and Δ*LAMA1*Enh cells. Data are shown as mean ± S.E.M. (n = 3 replicates from independent differentiations for control cells. Δ*LAMA1*Enh cells represent combined data from two clonal lines with three replicates for each line from independent differentiations. n = 3 technical replicates for each sample; p=0.319, 0.594, 0.945, 0.290, and $<1 \times 10^{-6}$ for comparisons in ES, DE, GT, PP1, and PP2, respectively; student's t-test, two sided; ***p<0.001, n.s., not significant). Light blue and purple indicate classification of the *LAMA1* PSSE as TE and SE, respectively. Each plotted point represents the average of technical replicates for each differentiation. (E) mRNA expression determined by RNA-seq at PP2 of genes expressed in either control or Δ*LAMA1*Enh cells (FPKM ≥ 1 at PP2) and located within the same topologically associated domain as *LAMA1*. Data are shown as mean FPKM ± S.E.M. (n = 2 replicates from independent differentiations for control cells. Δ*LAMA1*Enh cells represent combined data from two clonal lines with two replicates for each line from independent differentiations. p adj. = 0.389 and $8.11 \times 10^{-3}$ for *ARHGAP28* and *LAMA1*, respectively; DESeq2). See also *Figure 4—figure supplements 1* and *2*.

The online version of this article includes the following source data and figure supplement(s) for figure 4:

**Source data 1.** Genes downregulated in Δ*LAMA1*Enh PP2 stage cells compared to control cells (p adj. <0.05).
**Source data 2.** Genes upregulated in Δ*LAMA1*Enh PP2 stage cells compared to control cells (p adj. <0.05).
**Figure supplement 1.** Deletion of the *LAMA1*-associated pancreatic progenitor-specific enhancer does not affect pancreatic lineage specification.
**Figure supplement 2.** Deletion of *LAMA1* does not affect pancreatic lineage specification.

two independent clonal human hESC lines harboring homozygous deletions of the *LAMA1* PSSE (hereafter referred to as Δ*LAMA1*Enh; *Figure 4—figure supplement 1B*). We examined *LAMA1* expression in Δ*LAMA1*Enh compared to control cells throughout stages of pancreatic differentiation. Consistent with the broad expression of *LAMA1* across developmental and mature tissues, control cells expressed *LAMA1* at all stages (*Figure 4C*). *LAMA1* was expressed at similar levels in Δ*LAMA1*Enh and control cells at early developmental stages, but was significantly reduced in PP2 cells derived from Δ*LAMA1*Enh clones (p=0.319, 0.594, 0.945, 0.290, and $<1 \times 10^{-6}$ for comparisons in ES, DE, GT, PP1, and PP2, respectively; *Figure 4D*). To next investigate whether the *LAMA1* PSSE regulates other genes at this locus, we utilized Hi-C datasets from PP2 cells to identify topologically associated domains (TADs). We then examined expression of genes mapping in the same TAD as the *LAMA1* PSSE. *ARHGAP28* was the only other expressed gene within the TAD, and albeit not significantly different from controls (p.adj >0.05), showed a trend toward lower expression in Δ*LAMA1*Enh PP2 cells (*Figure 4E*), raising the possibility that *ARHGAP28* is an additional target gene of the *LAMA1* PSSE. Together, these results demonstrate that while *LAMA1* itself is broadly expressed across developmental stages, the T2D-associated PSSE regulates *LAMA1* expression specifically in pancreatic progenitors.

To determine whether deletion of the *LAMA1* PSSE affects pancreatic development, we generated PP2 stage cells from Δ*LAMA1*Enh and control hESC lines and analyzed pancreatic cell fate commitment by flow cytometry and immunofluorescence staining for PDX1 and NKX6.1 (*Figure 4—figure supplement 1C,D*). At the PP2 stage, Δ*LAMA1*Enh and control cultures contained similar percentages of PDX1- and NKX6.1-positive cells. Furthermore, mRNA expression of *PDX1*, *NKX6.1*, *PROX1*, *PTF1A*, and *SOX9* was either unaffected or only minimally reduced (p adj. = $3.56 \times 10^{-2}$, 0.224, 0.829, $8.14 \times 10^{-2}$, and 0.142, for comparisons of *PDX1*, *NKX6.1*, *PROX1*, *PTF1A*, and *SOX9* expression, respectively; *Figure 4—figure supplement 1E*), and the overall gene expression profiles as determined by RNA-seq were similar in Δ*LAMA1*Enh and control PP2 cells (*Figure 4—figure supplement 1F* and *Figure 4—source datas 1* and *2*). To examine effects of complete *LAMA1* loss-of-function, we additionally generated a hESC line harboring a deletion of the *LAMA1* coding sequences (hereafter referred to as Δ*LAMA1*; *Figure 4—figure supplement 2A,B*), and produced PP2 stage cells. Similar to Δ*LAMA1*Enh cultures, Δ*LAMA1* and control PP2 stage cultures contained similar numbers of PDX1- and NKX6.1-positive cells (*Figure 4—figure supplement 2C,D*). Likewise, mRNA expression of *PDX1*, *NKX6.1*, *PROX1*, *PTF1A*, and *SOX9* was similar in Δ*LAMA1* and control PP2 cells (p=$4.3 \times 10^{-2}$, 0.19, 0.16, 0.17, and $8.7 \times 10^{-2}$, respectively; *Figure 4—figure supplement 2E*). These findings indicate that in vitro pancreatic lineage induction is unperturbed in both

ΔLAMA1Enh cells exhibiting reduced *LAMA1* expression, as well as ΔLAMA1 cells where *LAMA1* coding sequences are disrupted.

## Pancreatic progenitor-specific stretch enhancers at the *CRB2* and *PGM1* loci harbor T2D-associated variants

Multiple variants with evidence for T2D association in PSSE mapped outside of known risk loci, such as those mapping to *CRB2* and *PGM1* (*Figure 3A*). As with the *LAMA1* PSSE, PSSE harboring variants at *CRB2* and *PGM1* were intronic to their respective genes, contained ATAC-seq peaks, and bound pancreatic lineage-determining transcription factors FOXA1, FOXA2, GATA4, GATA6, HNF6, SOX9, and PDX1 (*Figure 5A,B* and *Figure 5—figure supplement 1A,B*). Compared to the *LAMA1* PSSE, *CRB2* and *PGM1* PSSE were less specific to pancreatic progenitors and exhibited significant H3K27ac signal in several other tissues and cell types, most notably brain, liver, and the digestive tract (*Figure 5—figure supplement 1C,D*).

CRB2 is a component of the Crumbs protein complex involved in the regulation of cell polarity and neuronal, heart, retinal, and kidney development (*Alves et al., 2013*; *Bulgakova and Knust, 2009*; *Dudok et al., 2016*; *Jiménez-Amilburu and Stainier, 2019*; *Slavotinek et al., 2015*). However, its role in pancreatic development is unknown. To determine whether the *CRB2* PSSE regulates *CRB2* expression in pancreatic progenitors, we generated two independent hESC clones with homozygous deletions of the *CRB2* PSSE (hereafter referred to as ΔCRB2Enh; *Figure 5—figure supplement 2A*) and performed pancreatic differentiation of ΔCRB2Enh and control hESC lines. In control cells, *CRB2* was first expressed at the GT stage and increased markedly at the PP1 stage (*Figure 5C*). This pattern of *CRB2* expression is consistent with H3K27ac deposition at the *CRB2* PSSE in GT stage cells and classification as a SE at the PP1 and PP2 stages (*Figure 5A* and *Figure 5—figure supplement 1C*). In ΔCRB2Enh cells, we observed upregulation of *CRB2* expression at earlier developmental stages, in particular at the DE and GT stages ($p < 1 \times 10^{-6}$ at both stages; *Figure 5D*), suggesting that the *CRB2* PSSE may be associated with repressive transcriptional complexes prior to pancreas induction. At the PP2 stage, *CRB2* expression was significantly reduced in ΔCRB2Enh cells (p adj. = $3.51 \times 10^{-3}$; *Figure 5D*), whereas the expression of other genes in the same TAD was not affected (p adj. ≥0.05; *Figure 5E*). Thus, the *CRB2* PSSE specifically regulates *CRB2* and is required for *CRB2* expression in pancreatic progenitors.

Phenotypic characterization of PP2 stage ΔCRB2Enh cultures revealed similar percentages of PDX1- and NKX6.1-positive cells as in control cultures (*Figure 5—figure supplement 2B,C*). The expression of pancreatic transcription factors and global gene expression profiles were also similar in ΔCRB2Enh and control PP2 cells (*Figure 5—figure supplement 2D,E* and *Figure 5—source data 1*). Likewise, *CRB2* deletion hESCs (ΔCRB2) differentiated to the PP2 stage (*Figure 5—figure supplement 3A,B*) produced similar numbers of PDX1- and NKX6.1-positive cells and expressed pancreatic transcription factors at levels similar to control cells (*Figure 5—figure supplement 3C–E*). Thus, neither deletion of the *CRB2* PSSE nor the *CRB2* gene overtly impairs pancreatic lineage induction in the in vitro hESC differentiation system.

### *lama1* and *crb2* zebrafish morphants display annular pancreas and decreased beta cell mass

Based on their classification as extracellular matrix and cell polarity proteins, respectively, Laminin (encoded by *LAMA1*) and CRB2 are predicted to regulate processes related to tissue morphogenesis, such as cell migration, tissue growth, and cell allocation within the developing organ. Furthermore, PSSE in general were enriched for proximity to genes involved in tissue morphogenesis (*Figure 2E*), suggesting that T2D risk variants acting within PSSE could have roles in pancreas morphogenesis. Since cell migratory processes and niche-specific signaling events are not fully modeled during hESC differentiation, we reasoned that the in vitro pancreatic differentiation system might not be suitable for studying Laminin and CRB2 function in pancreatic development.

To circumvent these limitations, we employed zebrafish as an in vivo vertebrate model to study the effects of reduced *lama1* and *crb2* levels on pancreatic development. The basic organization and cell types in the pancreas as well as the genes regulating endocrine and exocrine pancreas development are highly conserved between zebrafish and mammals (*Dong et al., 2008*; *Field et al., 2003*; *Kimmel et al., 2015*). To analyze pancreatic expression of Laminin and Crb proteins, we used *Tg*

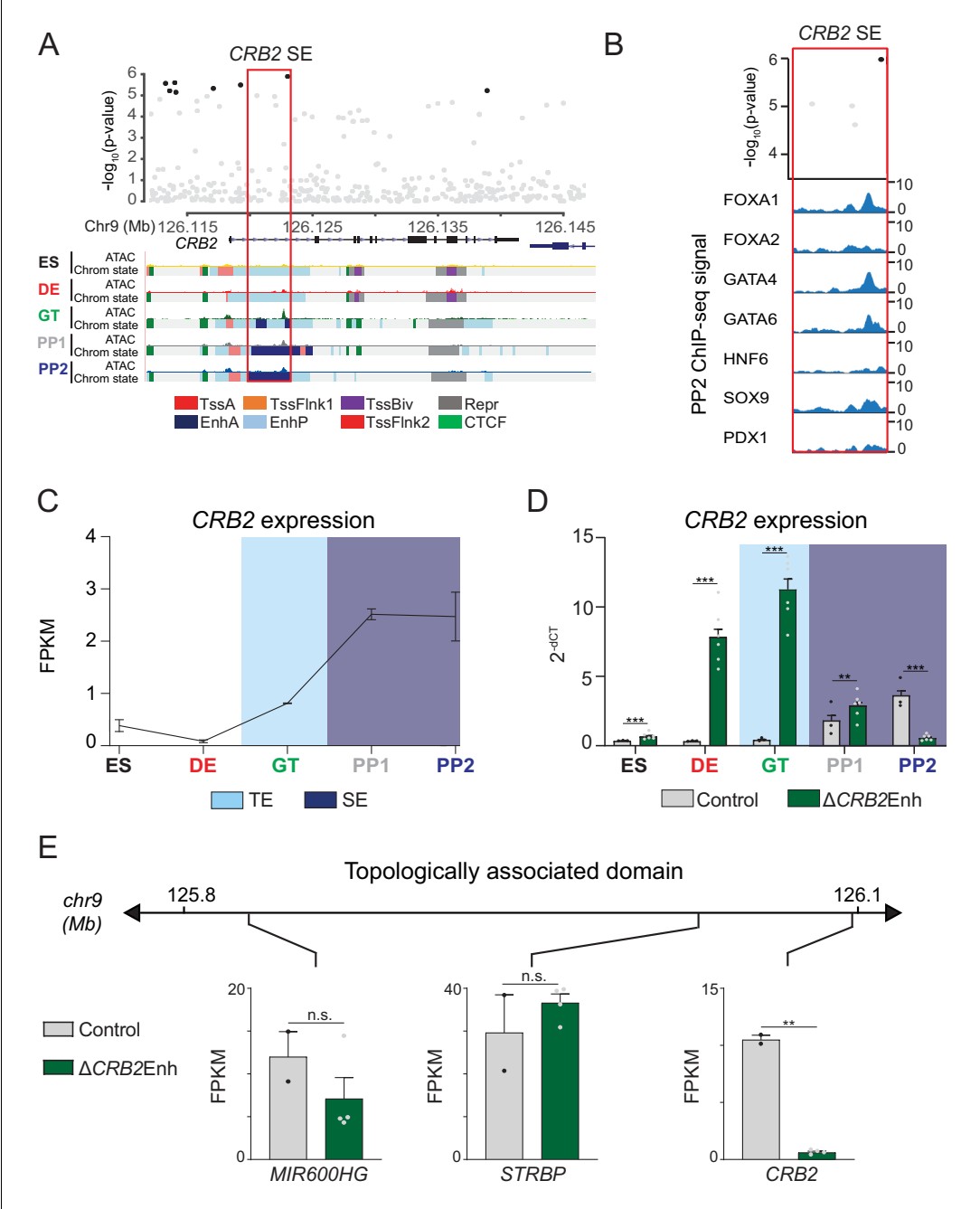

**Figure 5.** A T2D risk-associated *CRB2* pancreatic progenitor-specific stretch enhancer regulates *CRB2* expression specifically in pancreatic progenitors. (**A**) (Top) Locus plots showing T2D association p-values for variants in a 35 kb window (hg19 chr9:126,112,000–126,147,000) at the *CRB2* locus and *CRB2* PSSE (red box). Fine mapped variants within the 99% credible set for the novel *CRB2* locus are colored black. All other variants are colored light gray. (Bottom) Chromatin states and ATAC-seq signal in ES, DE, GT, PP1, and PP2. TssA, active promoter; TssFlnk, flanking transcription start site; TssBiv, bivalent promoter; Repr, repressed; EnhA, active enhancer; EnhP, poised enhancer. (**B**) FOXA1, FOXA2, GATA4, GATA6, HNF6, SOX9, and PDX1 ChIP-seq profiles at the *CRB2* PSSE in PP2. The variant rs2491353 (black) overlaps with transcription factor binding sites. (**C**) *CRB2* mRNA expression at each developmental stage determined by RNA-seq, measured in fragments per kilobase per million fragments mapped (FPKM). Data shown as mean ± S.E.M. (n = 3 replicates from independent differentiations). Light blue and purple indicate classification of the *CRB2* PSSE as typical enhancer (TE) and stretch enhancer (SE), respectively. Plotted points represent average of technical replicates for each differentiation. (**D**) *CRB2* mRNA expression at each developmental stage determined by qPCR in control and Δ*CRB2*Enh cells. Data are shown as mean ± S.E.M. (n = 3 replicates from independent differentiations for control cells. Δ*CRB2*Enh cells represent combined data from two clonal lines with three replicates for each line from independent differentiations. n = 3 technical replicates for each sample; p=7.03 × 10$^{-4}$,<1 × 10$^{-6}$,<1 × 10$^{-6}$, 1.46 × 10$^{-2}$, and <1 × 10$^{-6}$ for comparisons in ES, DE, GT, PP1, and PP2, respectively; student's t-test, two sided; ***p<0.001 **p<0.01). Light blue and purple indicate classification of the *CRB2* PSSE as TE

*Figure 5 continued on next page*

*Figure 5 continued*

and SE, respectively. Each plotted point represents the average of technical replicates for each differentiation. (E) mRNA expression determined by RNA-seq at PP2 of genes expressed in either control or ΔCRB2Enh cells (FPKM ≥ 1 at PP2) and located within the same topologically associated domain as *CRB2*. Data are shown as mean FPKM ± S.E.M. (n = 2 replicates from independent differentiations for control cells. ΔCRB2Enh cells represent combined data from two clonal lines with two replicates for each line from independent differentiations. p adj. = 0.158, 1.00, and $3.51 \times 10^{-3}$, for *MIR600HG*, *STRBP*, and *CRB2*, respectively; DESeq2; **p<0.01, n.s., not significant). See also *Figure 5—figure supplements 1–3*.

The online version of this article includes the following source data and figure supplement(s) for figure 5:

**Source data 1.** Genes downregulated in ΔCRB2Enh PP2 stage cells compared to control cells (p adj. <0.05).
**Figure supplement 1.** Activity of *CRB2*- and *PGM1*-associated pancreatic progenitor-specific stretch enhancers across human tissues.
**Figure supplement 2.** Deletion of the *CRB2*-associated pancreatic progenitor-specific enhancer does not affect pancreatic lineage specification.
**Figure supplement 3.** Deletion of *CRB2* does not affect pancreatic lineage specification.

*(ptf1a:eGFP)^{jh1}* embryos to visualize pancreatic progenitor cells and the acinar pancreas by eGFP expression. At 48 hr post-fertilization (hpf), both Laminin and Crb proteins were detected in the eGFP and Nkx6.1 co-positive pancreatic progenitor cell domain (*Figure 6—figure supplement 1A, B*).

To determine the respective functions of *lama1* and *crb2* in pancreatic development, we performed knockdown experiments using anti-sense morpholinos directed against *lama1* and the two zebrafish *crb2* genes, *crb2a* and *crb2b* (*Omori and Malicki, 2006*; *Pollard et al., 2006*). Knockdown efficiency of each morpholino was validated using whole-mount immunohistochemistry. We observed significant reduction of Laminin staining throughout the pancreatic progenitor cell domain in embryos treated with morpholinos targeting *lama1* (*Figure 6—figure supplement 2A–D*). In embryos treated with morpholinos targeting *crb2a* or *crb2a* and *crb2b*, we observed loss of staining in the pancreatic progenitor cell domain using antibodies specific to Crb2a or antibodies detecting all Crb proteins, respectively (*Figure 6—figure supplement 3A–H*) Residual panCrb protein signal was observed in the dorsal pancreas, which may be the result of expression of Crb proteins other than Crb2a and Crb2b in this region.

Consistent with prior studies (*Pollard et al., 2006*), *lama1* morphants exhibited reduced body size and other gross anatomical defects at 78 hpf, whereas *crb2a/b* morphants appeared grossly normal. Both *lama1* and *crb2a/b* morphants displayed an annular pancreas (15 out of 34 *lama1* and 27 out of 69 *crb2a/b* morphants) characterized by pancreatic tissue partially or completely encircling the duodenum (*Figure 6A–D*), a phenotype indicative of impaired migration of pancreatic progenitors during pancreas formation. These findings suggest that both *lama1* and *crb2a/b* control cell migratory processes during early pancreatic development and that reduced levels of *lama1* or *crb2a/b* impair pancreas morphogenesis.

To gain insight into the effects of *lama1* and *crb2a/b* knockdown on pancreatic endocrine cell development, we examined beta cell numbers (insulin$^+$ cells) at 78 hpf. We also evaluated potential synergistic effects of combined *lama1* and *crb2a/b* knockdown. To account for the reduction in body and pancreas size in *lama1* morphants, we compared cell numbers in 78 hpf *lama1* morphants with 50 hpf control embryos, which have a similarly sized acinar compartment as 78 hpf *lama1* morphants. Beta cell numbers were significantly reduced in both *lama1* and *crb2a/b* morphants (p=$8.0 \times 10^{-3}$ and $4.0 \times 10^{-3}$ for comparisons of *lama1* and *crb2a/b* morphants, respectively; *Figure 6E,F*), as well as in morphants with a combined knockdown of *lama1* and *crb2a/b* (p=$2.0 \times 10^{-4}$; *Figure 6F*), showing that reduced *lama1* and *crb2a/b* levels, both individually and in combination, impair beta cell development. Furthermore, we found that nearly all *lama1*, *crb2a/b*, and combined *lama1* and *crb2a/b* morphants without an annular pancreas had reduced beta cell numbers, indicating independent roles of *lama1* and *crb2* in pancreas morphogenesis and beta cell differentiation. Finally, to investigate the contributions of individual *crb2* genes to the observed phenotype, we performed knockdown experiments using morpholinos against *crb2a* and *crb2b* alone. Only *crb2b* morphants showed a significant reduction in beta cell numbers (p=$4.4 \times 10^{-2}$; *Figure 6—figure supplement 4*), suggesting that *crb2b* is the predominant *crb2* gene required for beta cell development. Combined, these findings demonstrate that *lama1* and *crb2* are regulators of pancreas morphogenesis and beta cell development in vivo.

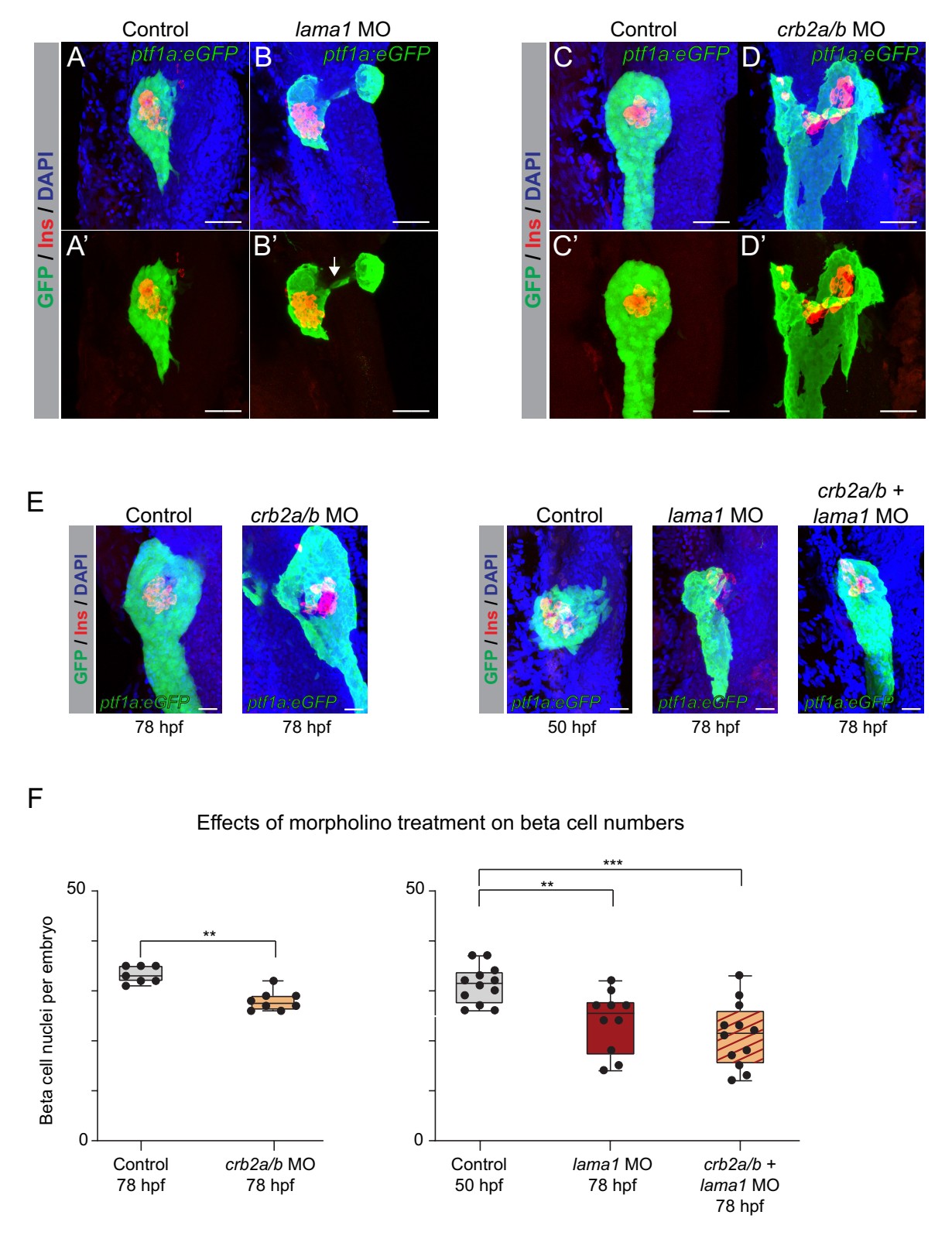

**Figure 6.** *lama1* and *crb2* regulate pancreas morphogenesis and beta cell differentiation. (**A,B**) Representative 3D renderings of *Tg(ptf1a: eGFP)^jh1* control zebrafish embryos (**A,A'**) and *lama1* morphants (**B,B'**) stained with DAPI (nuclei, blue) and antibody against insulin (red); n ≥ 15 embryos per condition. To account for reduced acinar pancreas size in *lama1* morphants, control embryos were imaged at 50 hr post fertilization (hpf) and *lama1* morphants at 78 hpf. 15 out of 34 *lama1* morphants displayed an annular pancreas with two acinar pancreas domains (green) connected

*Figure 6 continued on next page*

Figure 6 continued

behind the presumptive intestine (B', white arrow). Scale bar, 40 μM. (C,D) Representative 3D renderings of 78 hpf *Tg(ptf1a:eGFP)$^{jh1}$* control zebrafish embryos (C,C') and *crb2a/b* morphants (D,D') stained with DAPI (nuclei, blue) and antibodies against insulin (red); n ≥ 15 embryos per condition. Twenty-seven out of 69 *crb2a/b* morphants displayed an annular pancreas with the acinar pancreas (green) completely surrounding the presumptive intestine. Scale bar, 40 μM. (E) Representative 3D renderings of *Tg(ptf1a:eGFP)$^{jh1}$* control zebrafish embryos and *crb2a/b*, *lama1*, or *crb2a/b + lama1* morphants stained with DAPI (nuclei, blue) and antibody against insulin (red). All embryos were imaged at 78 hpf except for controls to *lama1* and *crb2a/b + lama1* morphants, which were imaged at 50 hpf to account for reduced acinar pancreas size of *lama1* morphants. Scale bar, 20 μM. (F) Quantification of beta (insulin$^+$) cell nuclei per embryo from experiment in (E). p adj. = $4.0 \times 10^{-3}$, $8.0 \times 10^{-3}$, and $2.0 \times 10^{-4}$ for comparison of hfp 78 control (n = 7 embryos) to hfp 78 *crb2a/b* (n = 8), hpf 50 control (n = 12) to hpf 78 *lama1* (n = 10), or *crb2a/b + lama1* (n = 12) morphants, respectively; ANOVA-Dunnett's multiple comparison test; ***p<0.001 **p<0.01. 5 out of 8 *crb2a/b*, 3 out of 10 *lama1*, and 9 out of 12 *crb2a/b + lama1* morphants displayed an annular pancreas. MO, morpholino; Control, standard control morpholino. See also *Figure 6—figure supplements 1–4*.

The online version of this article includes the following figure supplement(s) for figure 6:

**Figure supplement 1.** Laminin and Crb are expressed in zebrafish pancreas progenitors.
**Figure supplement 2.** Validation of morpholinos targeting *lama1*.
**Figure supplement 3.** Validation of morpholinos targeting *crb2a* and *crb2b*.
**Figure supplement 4.** *crb2b* but not *crb2a* regulates pancreatic beta cell differentiation.

## Discussion

In this study, we identify T2D-associated variants localized within chromatin active in pancreatic progenitors but not islets or other T2D-relevant tissues, suggesting a novel mechanism whereby a subset of T2D risk variants specifically alters pancreatic developmental processes. We link T2D-associated enhancers active in pancreatic progenitors to the regulation of *LAMA1* and *CRB2* and demonstrate a functional requirement in zebrafish for *lama1* and *crb2* in pancreas morphogenesis and endocrine cell formation. Furthermore, we provide a curated list of T2D risk-associated enhancers and candidate effector genes for further exploration of how the regulation of developmental processes in the pancreas can predispose to T2D.

Our analysis identified 11 loci where T2D-associated variants mapped in SE specifically active in pancreatic progenitors. Among these loci was *LAMA1*, which has stronger effects on T2D risk in lean compared to obese individuals (*Perry et al., 2012*). We also found evidence that variants in PSSE collectively have stronger enrichment for T2D in lean individuals, although the small number of PSSE and limited sample size of the BMI-stratified T2D genetic data prohibits a more robust comparison. There was also a notable lack of enrichment among PSSE variants for association with traits related to insulin secretion and beta cell function. If T2D-associated variants in PSSE indeed confer diabetes susceptibility by affecting beta cell development, the question arises as to why variants associated with traits related to beta cell function are not enriched within PSSE. As genetic association studies of endophenotypes are based on data from non-diabetic subjects, a possible explanation is that variants affecting beta cell developmental processes have no overt phenotypic effect under physiological conditions and contribute to T2D pathogenesis only during the disease process.

Since the genomic position of enhancers and transcription factor binding sites is not well conserved between species (*Villar et al., 2015*), a human cell model is necessary to identify target genes of enhancers associated with disease risk. By employing enhancer deletion in hESCs, we demonstrate that T2D-associated PSSE at the *LAMA1* and *CRB2* loci regulate *LAMA1* and *CRB2*, respectively, and establish *LAMA1* and *CRB2* as the predominant target gene of their corresponding PSSE within TAD boundaries. By analyzing *LAMA1* and *CRB2* expression throughout the pancreatic differentiation time course, we show that the identified PSSE control *LAMA1* and *CRB2* expression in a temporal manner consistent with the activation pattern of their associated PSSE. While the specific T2D-relevant target genes of the majority of T2D-associated PSSE remain to be identified, it is notable that several are localized within TADs containing genes encoding transcriptional regulators. These include *PROX1* and *GATA4*, which are known to regulate pancreatic development (*Shi et al., 2017*; *Tiyaboonchai et al., 2017*; *Westmoreland et al., 2012*), as well as *HMGA2* and *BCL6* with unknown functions in the pancreas. Our catalogue of T2D-associated PSSE provides a resource to fully characterize the gene regulatory program associated with developmentally mediated T2D risk in the pancreas. Our finding that predicted target genes of PSSE are similarly expressed in hESC-derived pancreatic progenitors and primary human embryonic pancreas (*Figure 3B* and *Figure 3— figure supplement 1A*) further underscores the utility of the hESC-based system for these studies.

In the embryo, endocrine cells differentiate by delaminating from a polarized epithelium of progenitors governed by local cell-cell and cell-matrix signaling events (*Mamidi et al., 2018*). These processes are not well-recapitulated in the hESC-based pancreatic differentiation system, highlighting a limitation of this system for studying the function of Laminin and CRB2, which are mediators of mechanical signals within an epithelium. Therefore, we analyzed their function in zebrafish as an in vivo model. We show that *lama1* or *crb2* knockdown leads to an annular pancreas and reduced beta cell numbers. The beta cell differentiation defect was also evident in embryos not displaying an annular pancreas, suggesting independent mechanisms.

Consistent with our findings in *lama1* morphants, culture of pancreatic progenitors on Laminin-based substrates promotes endocrine cell differentiation (*Mamidi et al., 2018*). During in vivo pancreatic development, endothelial cells are an important albeit not the only source of Laminin in the pancreas (*Heymans et al., 2019*; *Mamidi et al., 2018*; *Nikolova et al., 2006*). While we do not know the respective contributions of endothelial cell- and pancreatic progenitor cell-derived Laminin to the phenotype of *lama1* morphants, the T2D-associated *LAMA1* PSSE is not active in endothelial cells (*Figure 3—figure supplement 1C*). Furthermore, we found no other T2D-associated variants at the *LAMA1* locus mapping in endothelial cell enhancers or accessible chromatin sites in islets, suggesting that T2D risk is linked to *LAMA1* regulation in pancreatic progenitors.

Similar to Laminin, CRB2 has been shown to regulate mechanosignaling (*Varelas et al., 2010*). Our observation that pancreatic progenitor cells express Crb proteins is consistent with the phenotype of *crb2* morphants reflecting a progenitor-autonomous role of Crb2. Furthermore, the similarity in pancreatic phenotype between *lama1* or *crb2* morphants raises the possibility that signals from Laminin and Crb2 could converge on the same intracellular pathways in pancreatic progenitors.

Our findings suggest that variation in gene regulation during pancreatic development can predispose to T2D later in life. Several lines of evidence support the concept of a developmental impact on T2D risk. First, human genetic studies have shown a strong correlation between birth weight and adult cardiometabolic traits and disease (*Horikoshi et al., 2016*). Second, epidemiological studies provide evidence that offspring of mothers who were pregnant during a famine have a higher prevalence of T2D (*Lumey et al., 2015*). This phenomenon has been experimentally reproduced in rodents, where maternal malnutrition has been shown to cause reduced beta cell mass at birth and to render beta cells more prone to failure under stress (*Nielsen et al., 2014*). Together, our results provide a strong rationale for further exploration of how genetic variants affecting developmental gene regulation in the pancreas contribute to T2D risk.

# Materials and methods

## Key resources table

| Reagent type (species) or resource | Designation | Source or reference | Identifiers | Additional information |
|---|---|---|---|---|
| Antibody | APC Mouse monoclonal IgG1, κ Isotype Control | BD Pharmingen | Cat# 555751, RRID:AB_398613 | Flow cytometry (1:100) |
| Antibody | Chicken polyclonal anti-GFP | Aves Labs | Cat# GFP-1020, RRID:AB_10000240 | Immunohistochemistry (1:200) |
| Antibody | Cy3-conjugated donkey polyclonal anti-mouse | Jackson ImmunoResearch Labs | Cat# 715-165-150, RRID:AB_2340813 | Immunofluorescence (1:1000) |
| Antibody | DyLight 488-conjugated donkey polyconal anti-goat | Jackson ImmunoResearch Labs | Cat# 705-545-003, RRID:AB_2340428 | Immunofluorescence (1:500) |
| Antibody | Goat polyclonal anti-CTCF | Santa Cruz Biotechnology | Cat# SC-15914X, RRID:AB_2086899 | ChIP-seq (4 ug) |
| Antibody | Goat polyclonal anti-FOXA1 | Abcam | Cat# ab5089, RRID:AB_304744 | ChIP-seq (4 ug) |
| Antibody | Goat polyclonal anti-FOXA2 | Santa Cruz Biotechnology | Cat# sc-6554, RRID:AB_2262810 | ChIP-seq (4 ug) |

*Continued on next page*

*Continued*

| Reagent type (species) or resource | Designation | Source or reference | Identifiers | Additional information |
|---|---|---|---|---|
| Antibody | Goat polyclonal anti-GATA4 | Santa Cruz Biotechnology | Cat# sc-1237, RRID:AB_2108747 | ChIP-seq (4 ug) |
| Antibody | Goat polyclonal anti-PDX1 | Abcam | Cat# ab47383, RRID:AB_2162359 | Immunofluorescence (1:500) |
| Antibody | Guinea pig polyclonal anti-Insulin | Biomeda | Cat# v2024 | Immunohistochemistry (1:200) |
| Antibody | Mouse monoclonal anti-Crb2a | ZIRC | Cat# Zs-4 | Immunohistochemistry (1:100) |
| Antibody | Mouse polyclonal anti-GATA6 | Santa Cruz Biotechnology | Cat# sc-9055, RRID:AB_2108768 | ChIP-seq (4 ug) |
| Antibody | Mouse monoclonal anti-NKX6.1 | Developmental Studies Hybridoma Bank | Cat# F64A6B4, RRID:AB_532380 | Immunofluorescence (1:300) |
| Antibody | Mouse monoclonal anti-NKX6.1-Alexa Fluor 647 | BD Biosciences | Cat# 563338, RRID:AB_2738144 | Flow cytometry (1:5) |
| Antibody | Mouse monoclonal anti-NKX6.1 | Developmental Studies Hybridoma Bank | Cat# F55A10, RRID:AB_532378 | Immunohistochemistry (1:10) |
| Antibody | Mouse monoclonal anti-PDX1-PE | BD Biosciences | Cat# 562161, RRID:AB_10893589 | Flow cytometry (1:10) |
| Antibody | PE Mouse monoclonal IgG1, κ Isotype Control | BD Pharmingen | Cat# 555749, RRID:AB_396091 | Flow cytometry (1:100) |
| Antibody | Rabbit polyclonal anti-CRB2 | Sigma | Cat # SAB1301340 | Immunofluorescence (1:500) |
| Antibody | Rabbit polyclonal anti-H3K27ac | Active Motif | Cat# 39133, RRID:AB_2561016 | ChIP-seq (4 ug) |
| Antibody | Rabbit polyclonal anti-H3K4me1 | Abcam | Cat# ab8895, RRID:AB_306847 | ChIP-seq (4 ug) |
| Antibody | Rabbit polyclonal anti-HNF6 | Santa Cruz Biotechnology | Cat# sc-13050, RRID:AB_2251852 | ChIP-seq (4 ug) |
| Antibody | Rabbit polyclonal anti-laminin | Sigma | Cat# L9393, RRID:AB_477163 | Immunohistochemistry (1:100) Immunofluorescence (1:30) |
| Antibody | Rabbit monoclonal anti-panCrb | Jensen Laboratory, University of Massachusetts, Amherst | N/A | Immunohistochemistry (1:100) |
| Antibody | Rabbit polyclonal anti-PDX1 | Beta Cell Biology Consortium | AB1068 | ChIP-seq (4 ug) |
| Antibody | Rabbit polyclonal anti-SOX9 | Chemicon | Cat# 5535, RRID:AB_2239761 | ChIP-seq (4 ug) |
| Cell line (*Homo-sapiens*) | CyT49 | ViaCyte, Inc | NIHhESC-10–0041, RRID:CVCL_B850 | Male |
| Cell line (*Homo-sapiens*) | H1 | WiCell Research Institute | NIHhESC-10–0043, RRID:CVCL_9771 | Male |
| Chemical compound, drug | 2-Mercaptoethanol | Thermo Fisher Scientific | Cat# 21985023 | |
| Chemical compound, drug | Accutase | Thermo Fisher Scientific | Cat# 00-4555-56 | |
| Chemical compound, drug | B-27 supplement | Thermo Fisher Scientific | Cat# 17504044 | |

*Continued on next page*

Continued

| Reagent type (species) or resource | Designation | Source or reference | Identifiers | Additional information |
|---|---|---|---|---|
| Chemical compound, drug | Bovine Albumin Fraction V | Life Technologies | Cat# 15260037 | |
| Chemical compound, drug | D-(+)-Glucose Solution, 45% | Sigma-Aldrich | Cat# G8769 | |
| Chemical compound, drug | DAPI | Invitrogen | Cat# D1306 | Immunohistochemistry (1:200) |
| Chemical compound, drug | DMEM High Glucose | VWR | Cat# 16750–082 | |
| Chemical compound, drug | DMEM/F12 [-] L-glutamine | VWR | Cat# 15–090-CV | |
| Chemical compound, drug | DMEM/F12 with L-Glutamine, HEPES | Corning | Cat# 45000–350 | |
| Chemical compound, drug | DMF | EMD Millipore | Cat# DX1730 | |
| Chemical compound, drug | DPBS | Thermo Fisher Scientific | Cat# 21–031-CV | |
| Chemical compound, drug | DTT | Sigma | Cat# D9779 | |
| Chemical compound, drug | Fetal Bovine Serum | Thermo Fisher Scientific | Cat# MT35011CV | |
| Chemical compound, drug | Glutamax | Thermo Fisher Scientific | Cat# 35050–079 | |
| Chemical compound, drug | GlutaMAX | Thermo Fisher Scientific | Cat# 35050061 | |
| Chemical compound, drug | Hoechst 33342 | Thermo Fisher Scientific | Cat# H3570 | |
| Chemical compound, drug | HyClone Dulbecco's Modified Eagles Medium | Thermo Fisher Scientific | Cat# SH30081.FS | |
| Chemical compound, drug | IGEPAL-CA630 | Sigma | Cat# I8896 | |
| Chemical compound, drug | Illumina tagmentation enzyme | Illumina | Cat# FC-121–1030 | |
| Chemical compound, drug | Insulin-Transferrin-Selenium (ITS) | Thermo Fisher Scientific | Cat# 41400045 | |
| Chemical compound, drug | Insulin-Transferrin-Selenium-Ethanolamine (ITS-X) | Thermo Fisher Scientific | Cat# 51500–056 | |
| Chemical compound, drug | KAAD-Cyclopamine | Toronto Research Chemicals | Cat# K171000 | |
| Chemical compound, drug | K-acetate | Sigma | Cat# P5708 | |
| Chemical compound, drug | KnockOut SR XenoFree | Thermo Fisher Scientific | Cat# A1099202 | |
| Chemical compound, drug | LDN-193189 | Stemgent | Cat# 04–0074 | |
| Chemical compound, drug | Matrigel | Corning | Cat# 356231 | |
| Chemical compound, drug | MCDB 131 | Thermo Fisher Scientific | Cat# 10372–019 | |
| Chemical compound, drug | Mg-acetate | Sigma | Cat# M2545 | |
| Chemical compound, drug | mTeSR1 Complete Kit - GMP | STEMCELL Technologies | Cat# 85850 | |

*Continued*

| Reagent type (species) or resource | Designation | Source or reference | Identifiers | Additional information |
|---|---|---|---|---|
| Chemical compound, drug | NEBNext High-Fidelity 2X PCR Master Mix | NEB | Cat# M0541 | |
| Chemical compound, drug | Non-Essential Amino Acids | Thermo Fisher Scientific | Cat# 11140050 | |
| Chemical compound, drug | O.C.T. Compound | Sakura Finetek USA | Cat# 25608–930 | |
| Chemical compound, drug | Penicillin-Streptomycin | Thermo Fisher Scientific | Cat# 15140122 | |
| Chemical compound, drug | Polyethylenimine (PEI) | Polysciences | Cat# 23966–1 | |
| Chemical compound, drug | Protease inhibitor | Roche | Cat# 05056489001 | |
| Chemical compound, drug | Retinoic acid | Sigma-Aldrich | Cat# R2625 | |
| Chemical compound, drug | RNA ScreenTape Sample Buffer | Agilent Technologies | Cat# 5067–5577 | |
| Chemical compound, drug | ROCK Inhibitor Y-27632 | STEMCELL Technologies | Cat# 72305 | |
| Chemical compound, drug | RPMI 1640 [-] L-glutamine | VWR | Cat# 15–040-CV | |
| Chemical compound, drug | SANT-1 | Sigma-Aldrich | Cat# S4572 | |
| Chemical compound, drug | Sodium Bicarbonate | Sigma-Aldrich | Cat# NC0564699 | |
| Chemical compound, drug | Tamoxifen | Sigma | Cat# T5648 | |
| Chemical compound, drug | TGF-β RI Kinase Inhibitor IV | Calbiochem | Cat# 616454 | |
| Chemical compound, drug | TPB | Calbiochem | Cat# 565740 | |
| Chemical compound, drug | Tranylcypromine | Cayman Chemical | Cat# 10010494 | |
| Chemical compound, drug | Tris-acetate | Thermo Fisher Scientific | Cat# BP-152 | |
| Chemical compound, drug | TTNPB | Enzo Life Sciences | Cat# BML-GR105 | |
| Chemical compound, drug | Vectashield Antifade Mounting Medium | Vector Laboratories | Cat# H-1000 | |
| Chemical compound, drug | XtremeGene 9 | Roche | Cat# 6365787001 | |
| Commercial assay | High Sensitivity D1000 ScreenTape | Agilent Technologies | Cat# 5067–5584 | |
| Commercial assay, kit | RNA ScreenTape | Agilent Technologies | Cat# 5067–5576 | |
| Commercial assay, kit | RNA Screen Tape Ladder | Agilent Technologies | Cat# 5067–5578 | |
| Commercial assay, kit | BD Cytofix/ Cytoperm Plus Fixation/ Permeabilization Solution Kit | BD Biosciences | Cat# 554715 | |
| Commercial assay, kit | ChIP-IT High Sensitivity Kit | Active Motif | Cat# 53040 | |

*Continued on next page*

*Continued*

| Reagent type (species) or resource | Designation | Source or reference | Identifiers | Additional information |
|---|---|---|---|---|
| Commercial assay, kit | iQ SYBR Green Supermix | Bio-Rad | Cat# 1708884 | |
| Commercial assay, kit | iScript cDNA Synthesis Kit | Bio-Rad | Cat# 1708891 | |
| Commercial assay, kit | KAPA Library Preparation Kit (Illumina) | Kapa Biosystems | Cat# KK8234 | |
| Commercial assay, kit | KAPA Stranded mRNA-Seq Kits | Kapa Biosystems | Cat# KK8401 | |
| Commercial assay, kit | MinElute PCR purification kit | QIAGEN | Cat# 28004 | |
| Commercial assay, kit | Qubit ssDNA assay kit | Thermo Fisher Scientific | Cat# Q10212 | |
| Commercial assay, kit | RNeasy Micro Kit | QIAGEN | Cat# 74004 | |
| Genetic reagent (*D. rerio*) | *Tg(ptf1a:eGFP)*[jh1] | PMID:16258076 | N/A | |
| Other | SPRIselect bead | Beckman Coulter | Cat# B23317 | |
| Recombinant protein | Activin A | R and D Systems | Cat# 338-AC/CF | |
| Recombinant protein | Human AB Serum | Valley Biomedical | Cat# HP1022 | |
| Recombinant protein | Recombinant EGF | R and D Systems | Cat# 236-EG | |
| Recombinant protein | Recombinant Heregulinβ−1 | Peprotech | Cat# 100–03 | |
| Recombinant protein | Recombinant KGF/FGF7 | R and D Systems | Cat# 251 KG | |
| Recombinant protein | Recombinant Mouse Wnt3A | R and D Systems | Cat# 1324-WN/CF | |
| Recombinant protein | Recombinant Noggin | R and D Systems | Cat# 3344 NG | |
| Sequence-based reagent | Px333 Plasmid | http://www.addgene.org/64073/ | RRID:Addgene_64073 | |
| Sequence-based reagent | LAMA1 Forward | This paper | qPCR primers | GTG ATG GCA ACA GCG CAA A |
| Sequence-based reagent | LAMA1 Reverse | This paper | qPCR primers | GAC CCA GTG ATA TTC TCT CCC A |
| Sequence-based reagent | CRB2 Forward | This paper | qPCR primers | ACC ACT GTG CTT GTC CTG AG |
| Sequence-based reagent | CRB2 Reverse | This paper | qPCR primers | TCC AGG GTC GCT AGA TGG AG |
| Sequence-based reagent | TBP Forward | This paper | qPCR primers | TGT GCA CAG GAG CCA AGA GT |
| Sequence-based reagent | TBP Reverse | This paper | qPCR primers | ATT TTC TTG CTG CCA GTC TGG |
| Sequence-based reagent | *LAMA1*Enh Upstream Guide | This paper | CRISPR sgRNA | GTC AAA TTG CTA TAA CAC GG |
| Sequence-based reagent | *LAMA1*Enh Downstream Guide | This paper | CRISPR sgRNA | CCA CTT TAA GTA TCT CAG CA |
| Sequence-based reagent | *CRB2*Enh Upstream Guide | This paper | CRISPR sgRNA | ATA CAA AGC ACG TGA GA |

*Continued on next page*

*Continued*

| Reagent type (species) or resource | Designation | Source or reference | Identifiers | Additional information |
|---|---|---|---|---|
| Sequence-based reagent | *CRB2*Enh Downstream Guide | This paper | CRISPR sgRNA | GAA TGC GGA TGA CGC CTG AG |
| Sequence-based reagent | lama1-ATG | PMID:16321372 | Morpholino | TCA TCC TCA TCT CCA TCA TCG CTC A<br>Obtained from GeneTools, LLC |
| Sequence-based reagent | crb2a-SP | PMID:16713951 | Morpholino | ACG TTG CCA GTA CCT GTG TAT CCT G<br>Obtained from GeneTools, LLC |
| Sequence-based reagent | crb2b-SP | PMID:16713951 | Morpholino | TAA AGA TGT CCT ACC CAG CTT GAA C<br>Obtained from GeneTools, LLC |
| Sequence-based reagent | standard control MO | N/A | Morpholino | CCT CTT ACC TCA GTT ACA ATT TAT A<br>Obtained from GeneTools, LLC |
| Software, algorithm | Adobe Illustrator v 5.1 | http://www.adobe.com/products/illustrator.html | RRID:SCR_014198 | |
| Software, algorithm | Adobe Photoshop v 5.1 | http://www.adobe.com/products/photoshop.html | RRID:SCR_014199 | |
| Software, algorithm | BEDtools v 2.26.0 | https://github.com/arq5x/bedtools2 | RRID:SCR_006646 | |
| Software, algorithm | Bioconductor | https://www.bioconductor.org/ | RRID: SCR_006442 | |
| Software, algorithm | Burrows-Wheeler Aligner v 0.7.13 | http://bio-bwa.sourceforge.net/ | RRID:SCR_010910 | |
| Software, algorithm | CENTIPEDE v 1.2 | http://centipede.uchicago.edu/ | N/A | |
| Software, algorithm | Cufflinks v 2.2.1 | http://cole-trapnell-lab.github.io/cufflinks/ | RRID:SCR_014597 | |
| Software, algorithm | deepTools2 v 3.1.3 | https://deeptools.readthedocs.io/en/develop/content/installation.html | N/A | |
| Software, algorithm | DESeq2 v 3.10 | https://bioconductor.org/packages/release/bioc/html/DESeq2.html | RRID:SCR_015687 | |
| Software, algorithm | FlowJo v10 software | https://www.flowjo.com/solutions/flowjo | RRID: SCR_008520 | |
| Software, algorithm | GraphPad Prism v 8.1.2 | https://www.graphpad.com/scientific-software/prism/ | RRID: SCR_002798 | |
| Software, algorithm | HOMER v 4.10.4 | http://homer.ucsd.edu/homer/ | RRID: SCR_010881 | |
| Software, algorithm | Juicebox Tools v 1.4 | https://github.com/aidenlab/Juicebox/wiki/Juicebox-Assembly-Tools | N/A | |
| Software, algorithm | MACS2 v 2.1.4 | http://liulab.dfci.harvard.edu/MACS/ | RRID:SCR_013291 | |
| Software, algorithm | MEME suite v 5.1.1 | http://meme-suite.org/ | RRID:SCR_001783 | |
| Software, algorithm | Metascape | http://metscape.ncibi.org | RRID:SCR_014687 | |
| Software, algorithm | Picard Tools v 1.131 | http://broadinstitute.github.io/picard/ | RRID:SCR_006525 | |
| Software, algorithm | R Project for Statistical Computing v 3.6.1 | http://www.r-project.org/ | RRID:SCR_001905 | |
| Software, algorithm | SAMtools v 1.5 | http://samtools.sourceforge.net | RRID:SCR_002105 | |
| Software, algorithm | STAR v 2.4 | https://github.com/alexdobin/STAR | N/A | |
| Software, algorithm | UCSC Genome Browser | http://genome.ucsc.edu/ | RRID:SCR_005780 | |
| Software, algorithm | vcf2diploid v 0.2.6a | https://github.com/abyzovlab/vcf2diploid | N/A | |
| Software, algorithm | ZEISS ZEN Digital Imaging for Light Microscopy | http://www.zeiss.com/microscopy/en_us/products/microscope-software/zen.html#introduction | RRID:SCR_013672 | |

## Maintenance and differentiation of CyT49 hESCs

Genomic and gene expression analyses (ChIP-seq, ATAC-seq, RNA-seq) for generation of chromatin maps and target gene identification were performed in CyT49 hESCs (male). Propagation of CyT49 hESCs was carried out by passing cells every 3 to 4 days using Accutase (eBioscience) for enzymatic cell dissociation, and with 10% (v/v) human AB serum (Valley Biomedical) included in the hESC media the day of passage. hESCs were seeded into tissue culture flasks at a density of 50,000 cells/cm$^2$. hESC research was approved by the University of California, San Diego, Institutional Review Board and Embryonic Stem Cell Research oversight committee.

Pancreatic differentiation was performed as previously described (*Schulz et al., 2012*; *Wang et al., 2015*; *Xie et al., 2013*). Briefly, a suspension-based culture format was used to differentiate cells in aggregate form. Undifferentiated aggregates of hESCs were formed by re-suspending dissociated cells in hESC maintenance medium at a concentration of $1 \times 10^6$ cells/mL and plating 5.5 mL per well of the cell suspension in 6-well ultra-low attachment plates (Costar). The cells were cultured overnight on an orbital rotator (Innova2000, New Brunswick Scientific) at 95 rpm. After 24 hr the undifferentiated aggregates were washed once with RPMI medium and supplied with 5.5 mL of day 0 differentiation medium. Thereafter, cells were supplied with the fresh medium for the appropriate day of differentiation (see below). Cells were continually rotated at 95 rpm, or 105 rpm on days 4 through 8, and no media change was performed on day 10. Both RPMI (Mediatech) and DMEM High Glucose (HyClone) medium were supplemented with 1X GlutaMAX and 1% penicillin/streptomycin. Human activin A, mouse Wnt3a, human KGF, human noggin, and human EGF were purchased from R and D systems. Other added components included FBS (HyClone), B-27 supplement (Life Technologies), Insulin-Transferrin-Selenium (ITS; Life Technologies), TGFβ R1 kinase inhibitor IV (EMD Bioscience), KAAD-Cyclopamine (KC; Toronto Research Chemicals), and the retinoic receptor agonist TTNPB (RA; Sigma Aldrich). Day-specific differentiation media formulations were as follows:

> Days 0 and 1: RPMI + 0.2% (v/v) FBS, 100 ng/mL Activin, 50 ng/mL mouse Wnt3a, 1:5000 ITS.
> Days 1 and 2: RPMI + 0.2% (v/v) FBS, 100 ng/mL Activin, 1:5000 ITS
> Days 2 and 3: RPMI + 0.2% (v/v) FBS, 2.5 mM TGFβ R1 kinase inhibitor IV, 25 ng/mL KGF, 1:1000 ITS
> Days 3–5: RPMI + 0.2% (v/v) FBS, 25 ng/mL KGF, 1:1000 ITS
> Days 5–8: DMEM + 0.5X B-27 Supplement, 3 nM TTNPB, 0.25 mM KAAD-Cyclopamine, 50 ng/mL Noggin
> Days 8–10: DMEM/B-27, 50 ng/mL KGF, 50 ng/mL EGF

Cells at D0 correspond to the embryonic stem cell (ES) stage, cells at D2 correspond to the definitive endoderm (DE) stage, cells at D5 correspond to the gut tube (GT) stage, cells at D7 correspond to the early pancreatic progenitor (PP1) stage, and cells at D10 correspond to the late pancreatic progenitor (PP2) stage.

## Maintenance and differentiation of H1 hESCs

ΔLAMA1Enh and ΔCRB2Enh clonal lines were derived by targeting H1 hESCs (male). Cells were maintained and differentiated as described with some modifications (*Jin et al., 2019*; *Rezania et al., 2014*). In brief, hESCs were cultured in mTeSR1 media (Stem Cell Technologies) and propagated by passaging cells every 3–4 days using Accutase (eBioscience) for enzymatic cell dissociation. hESC research was approved by the University of California, San Diego, Institutional Review Board and Embryonic Stem Cell Research Oversight Committee.

For differentiation, cells were dissociated using Accutase for 10 min, then reaggregated by plating the cells at a concentration of ~5.5 e6 cells/well in a low attachment six-well plate on an orbital shaker (100 rpm) in a 37°C incubator. The following day, undifferentiated cells were washed in base media (see below) and then differentiated using a multi-step protocol with stage-specific media and daily media changes.

All stage-specific base media were comprised of MCDB 131 medium (Thermo Fisher Scientific) supplemented with NaHCO3, GlutaMAX, D-Glucose, and BSA using the following concentrations:

Stage 1/2 medium: MCDB 131 medium, 1.5 g/L NaHCO3, 1X GlutaMAX, 10 mM D-Glucose, 0.5% BSA
Stage 3/4 medium: MCDB 131 medium, 2.5 g/L NaHCO3, 1X GlutaMAX, 10 mM D-glucose, 2% BSA

Media compositions for each stage were as follows:

Stage 1 (day 0–2): base medium, 100 ng/ml Activin A, 25 ng/ml Wnt3a (day 0). Day 1–2: base medium, 100 ng/ml Activin A
Stage 2 (day 3–5): base medium, 0.25 mM L-Ascorbic Acid (Vitamin C), 50 ng/mL FGF7
Stage 3 (day 6–7): base medium, 0.25 mM L-Ascorbic Acid, 50 ng/mL FGF7, 0.25 µM SANT-1, 1 µM Retinoic Acid, 100 nM LDN193189, 1:200 ITS-X, 200 nM TPB
Stage 4 (day 8–10): base medium, 0.25 mM L-Ascorbic Acid, 2 ng/mL FGF7, 0.25 µM SANT-1, 0.1 µM Retinoic Acid, 200 nM LDN193189, 1:200 ITS-X, 100 nM TPB

Cells at D0 correspond to the embryonic stem cell (ES) stage, cells at D3 correspond to the definitive endoderm (DE) stage, cells at D6 correspond to the gut tube (GT) stage, cells at D8 correspond to the early pancreatic progenitor (PP1) stage, and cells at D11 correspond to the late pancreatic progenitor (PP2) stage.

## Generation of ∆*LAMA1*Enh, ∆*CRB2*Enh, ∆*LAMA1*, and ∆*CRB2* hESC lines

To generate clonal homozygous *LAMA1*Enh and *CRB2*Enh deletion hESC lines, sgRNAs targeting each relevant enhancer were designed and cloned into Px333-GFP, a modified version of Px333 (Addgene, #64073). To generate clonal homozygous *LAMA1* and *CRB2* deletion hESC lines, sgRNAs targeting the second exon of each gene were designed and cloned into Px458 (Addgene, #48138). Plasmids expressing the sgRNAs were transfected into H1 hESCs with XtremeGene 9 (Roche). Twenty-four hr later, 8000 GFP+ cells were sorted into a well of six-well plate. Individual colonies that emerged within 5–7 days after transfection were subsequently transferred manually into 48-well plates for expansion, genomic DNA extraction, PCR genotyping, and Sanger sequencing. sgRNA oligos are listed below.

*LAMA1*Enh Upstream Guide: GTCAAATTGCTATAACACGG
*LAMA1*Enh Downstream Guide: CCACTTTAAGTATCTCAGCA
*CRB2*Enh Upstream Guide: ATACAAAGCACGTGAGA
*CRB2*Enh Downstream Guide: GAATGCGGATGACGCCTGAG
*LAMA1* Exon 2 Guide: ATCAGCACCAATGCCACCTG
*CRB2* Exon 2 Guide: TCGATGTCCAGCTCGCAGCG

## Human tissue

Human embryonic pancreas tissue was obtained from the Birth Defects Research Laboratory of the University of Washington. Studies for use of embryonic human tissue were approved by the Institutional Review Board of the University of California, San Diego. A pancreas from a 54- and 58-day gestation embryo each were pooled for RNA-seq analysis.

## Zebrafish husbandry

Adult zebrafish and embryos were cared for and maintained under standard conditions. All research activity involving zebrafish was reviewed and approved by SBP Medical Discovery Institute Institutional Animal Care and Use Committee. The following transgenic lines were used: *Tg(ptf1a:eGFP)*[jh1] (*Godinho et al., 2005*).

## Morpholino injections in zebrafish

The following previously validated morpholinos were injected into the yolk at the one-cell stage in a final volume of either 0.5 or 1 nl: 0.75 ng lama1-ATG (5'- TCATCCT CATCTCCATCATCGCTCA −3'); 3 ng crb2a-SP, (5'-ACGTTGCCAGTACCTGTGTATCCTG-3') (*Omori and Malicki, 2006*; *Watanabe et al., 2010*); 3 ng crb2b-SP, (5'-TAAAGATGTCCTACCCAGCTTGAAC-3') (*Omori and Malicki, 2006*); 6.75 ng standard control MO (5'- CCTCTTACCTCAGTTACAATTTATA −3'). All morpholinos were obtained from GeneTools, LLC.

## Chromatin immunoprecipitation sequencing (ChIP-seq)

ChIP-seq was performed using the ChIP-IT High-Sensitivity kit (Active Motif) according to the manufacturer's instructions. Briefly, for each cell stage and condition analyzed, $5–10 \times 10^6$ cells were harvested and fixed for 15 min in an 11.1% formaldehyde solution. Cells were lysed and homogenized using a Dounce homogenizer and the lysate was sonicated in a Bioruptor Plus (Diagenode), on high for $3 \times 5$ min (30 s on, 30 s off). Between 10 and 30 µg of the resulting sheared chromatin was used for each immunoprecipitation. Equal quantities of sheared chromatin from each sample were used for immunoprecipitations carried out at the same time. A total of 4 µg of antibody were used for each ChIP-seq assay. Chromatin was incubated with primary antibodies overnight at 4°C on a rotator followed by incubation with Protein G agarose beads for 3 hr at 4°C on a rotator. Antibodies used were rabbit anti-H3K27ac (Active Motif 39133), rabbit anti-H3K4me1 (Abcam ab8895), rabbit anti-H3K4me3 (Millipore 04–745), rabbit anti-H3K27me3 (Millipore 07–499), goat anti-CTCF (Santa Cruz Biotechnology SC-15914X), goat anti-GATA4 (Santa Cruz SC-1237), rabbit anti-GATA6 (Santa Cruz SC-9055), goat anti-FOXA1 (Abcam Ab5089), goat-anti-FOXA2 (Santa Cruz SC-6554), rabbit anti-PDX1 (BCBC AB1068), rabbit anti-HNF6 (Santa Cruz SC-13050), and rabbit anti-SOX9 (Chemicon AB5535). Reversal of crosslinks and DNA purification were performed according to the ChIP-IT High-Sensitivity instructions, with the modification of incubation at 65°C for 2–3 hr, rather than at 80°C for 2 hr. Sequencing libraries were constructed using KAPA DNA Library Preparation Kits for Illumina (Kapa Biosystems) and library sequencing was performed on either a HiSeq 4000 System (Illumina) or NovaSeq 6000 System (Illumina) with single-end reads of either 50 or 75 base pairs (bp). Sequencing was performed by the Institute for Genomic Medicine (IGM) core research facility at the University of California at San Diego (UCSD). Two replicates from independent hESC differentiations were generated for each ChIP-seq experiment.

## ChIP-seq data analysis

ChIP-seq reads were mapped to the human genome consensus build (hg19/GRCh37) and visualized using the UCSC Genome Browser (*Kent et al., 2002*). Burrows-Wheeler Aligner (BWA) (*Li and Durbin, 2009*) version 0.7.13 was used to map data to the genome. Unmapped and low-quality (q < 15) reads were discarded. SAMtools (*Li et al., 2009*) was used to remove duplicate sequences and HOMER (*Heinz et al., 2010*) was used to call peaks using default parameters and to generate tag density plots. Stage- and condition-matched input DNA controls were used as background when calling peaks. The BEDtools suite of programs (*Quinlan and Hall, 2010*) was used to perform genomic algebra operations. For all ChIP-seq experiments, replicates from two independent hESC differentiations were generated. Tag directories were created for each replicate using HOMER. Directories from each replicate were then combined, and peaks were called from the combined replicates. For histone modifications and CTCF peaks, pearson correlations between each pair of replicates were calculated over the called peaks using the command multiBamSummary from the deepTools2 package (*Ramírez et al., 2016*). For pancreatic lineage-determining transcription factors (GATA4, GATA6, FOXA1, FOXA2, HNF6, PDX1, SOX9), correlations were calculated for peaks overlapping PSSE. Calculated Pearson correlations are as follow:

|     | H3K4me1 | H3K27ac | CTCF | H3K4me3 | H3K27me3 |
| --- | --- | --- | --- | --- | --- |
| ES  | 0.90 | 0.91 | 0.87 | 0.81 | 1.00 |
| DE  | 0.97 | 0.84 | 0.86 | 0.99 | 0.99 |
| GT  | 0.97 | 0.87 | 0.89 | 0.97 | 0.99 |
| PP1 | 0.97 | 0.85 | 0.89 | 0.96 | 0.99 |
| PP2 | 0.98 | 0.87 | 0.87 | 0.97 | 1.00 |

|     | GATA4 | GATA6 | FOXA1 | FOXA2 | HNF6 | PDX1 | SOX9 |
| --- | --- | --- | --- | --- | --- | --- | --- |
| PP2 | 0.86 | 0.82 | 0.87 | 0.80 | 0.95 | 0.64 | 0.86 |

## RNA isolation and sequencing (RNA-seq) and qRT-PCR

RNA was isolated from cell samples using the RNeasy Micro Kit (Qiagen) according to the manufacturer instructions. For each cell stage and condition analyzed between 0.1 and $1 \times 10^6$ cells were collected for RNA extraction. For qRT-PCR, cDNA synthesis was first performed using the iScript cDNA Synthesis Kit (Bio-Rad) and 500 ng of isolated RNA per reaction. qRT-PCR reactions were performed in triplicate with 10 ng of template cDNA per reaction using a CFX96 Real-Time PCR Detection System and the iQ SYBR Green Supermix (Bio-Rad). PCR of the TATA binding protein (TBP) coding sequence was used as an internal control and relative expression was quantified via double delta CT analysis. For RNA-seq, stranded, single-end sequencing libraries were constructed from isolated RNA using the TruSeq Stranded mRNA Library Prep Kit (Illumina) and library sequencing was performed on either a HiSeq 4000 System (Illumina) or NovaSeq 6000 System (Illumina) with single-end reads of either 50 or 75 base pairs (bp). Sequencing was performed by the Institute for Genomic Medicine (IGM) core research facility at the University of California at San Diego. A complete list of RT-qPCR primer sequences can be found below.

| *LAMA1* forward | GTG ATG GCA ACA GCG CAA A |
| --- | --- |
| *LAMA1* reverse | GAC CCA GTG ATA TTC TCT CCC A |
| *CRB2* forward | ACC ACT GTG CTT GTC CTG AG |
| *CRB2* reverse | TCC AGG GTC GCT AGA TGG AG |
| *PDX1* forward | AAG TCT ACC AAA GCT CAC GCG |
| *PDX1* reverse | GTA GGC GCC GCC TGC |
| *NKX6.1* forward | CTG GCC TGT ACC CCT CAT CA |
| *NKX6.1* reverse | CTT CCC GTC TTT GTC CAA CA |
| *PROX1* forward | AAC ATG CAC TAC AAT AAA GCA AAT GAC |
| *PROX1* reverse | CAG GAA TCT CTC TGG AAC CTC AAA |
| *PTF1A* forward | GAA GGT CAT CAT CTG CCA TC |
| *PTF1A* reverse | GGC CAT AAT CAG GGT CGC T |
| *SOX9* forward | AGT ACC CGC ACT TGC ACA AC |
| *SOX9* reverse | ACT TGT AAT CCG GGT GGT CCT T |
| *TBP* forward | TGT GCA CAG GAG CCA AGA GT |
| *TBP* reverse | ATT TTC TTG CTG CCA GTC TGG |

## RNA-seq data analysis

Reads were mapped to the human genome consensus build (hg19/GRCh37) using the Spliced Transcripts Alignment to a Reference (STAR) aligner v2.4 (*Dobin et al., 2013*). Normalized gene expression (fragments per kilobase per million mapped reads; FPKM) for each sequence file was determined using Cufflinks v2.2.1 (*Trapnell et al., 2010*) with the parameters: `–library-type` fr-firststrand `–max-bundle-frags` 10000000. For all RNA-Seq experiments, replicates from two independent hESC differentiations were generated. Pearson correlations between bam files corresponding to each pair of replicates were calculated, and are as follow:

| Δ*LAMA1*Enh clone 1 PP2 | 1.00 |
| --- | --- |
| Δ*LAMA1*Enh clone 2 PP2 | 0.99 |
| Δ*CRB2*Enh clone 1 PP2 | 0.98 |
| Δ*CRB2*Enh clone 2 PP2 | 0.90 |
| Δ*LAMA1*Enh control PP2 | 0.92 |
| Δ*CRB2*Enh control PP2 | 0.99 |

## Assay for transposase accessible chromatin sequencing (ATAC-seq)

ATAC-seq (*Buenrostro et al., 2013*) was performed on approximately 50,000 nuclei. The samples were permeabilized in cold permeabilization buffer 0.2% IGEPAL-CA630 (I8896, Sigma), 1 mM DTT (D9779, Sigma), Protease inhibitor (05056489001, Roche), 5% BSA (A7906, Sigma) in PBS (10010–23, Thermo Fisher Scientific) for 10 min on the rotator in the cold room and centrifuged for 5 min at 500 × g at 4°C. The pellet was resuspended in cold tagmentation buffer (33 mM Tris-acetate (pH = 7.8) (BP-152, Thermo Fisher Scientific), 66 mM K-acetate (P5708, Sigma), 11 mM Mg-acetate (M2545, Sigma), 16% DMF (DX1730, EMD Millipore) in Molecular biology water (46000 CM, Corning)) and incubated with tagmentation enzyme (FC-121–1030; Illumina) at 37°C for 30 min with shaking at 500 rpm. The tagmented DNA was purified using MinElute PCR purification kit (28004, QIAGEN). Libraries were amplified using NEBNext High-Fidelity 2X PCR Master Mix (M0541, NEB) with primer extension at 72°C for 5 min, denaturation at 98°C for 30 s, followed by 8 cycles of denaturation at 98°C for 10 s, annealing at 63°C for 30 s and extension at 72°C for 60 s. After the purification of amplified libraries using MinElute PCR purification kit (28004, QIAGEN), double size selection was performed using SPRIselect bead (B23317, Beckman Coulter) with 0.55X beads and 1.5X to sample volume. Finally, libraries were sequenced on HiSeq4000 (Paired-end 50 cycles, Illumina).

## ATAC-seq data analysis

ATAC-seq reads were mapped to the human genome (hg19/GRCh37) using Burrows-Wheeler Aligner (BWA) version 0.7.13 (*Li and Durbin, 2009*), and visualized using the UCSC Genome Browser (*Kent et al., 2002*). SAMtools (*Li et al., 2009*) was used to remove unmapped, low-quality ($q < 15$), and duplicate reads. MACS2 (*Zhang et al., 2008*) was used to call peaks, with parameters 'shift set to 100 bps, smoothing window of 200 bps' and with 'nolambda' and 'nomodel' flags on. MACS2 was also used to call ATAC-Seq summits, using the same parameters combined with the 'call-summits' flag.

For all ATAC-Seq experiments, replicates from two independent hESC differentiations were generated. Bam files for each pair of replicates were merged for downstream analysis using SAMtools, and Pearson correlations between bam files for each individual replicate were calculated over a set of peaks called from the merged bam file. Correlations were performed using the command multiBamSummary from the deepTools2 package (*Ramírez et al., 2016*) with the '—removeOutliers' flag and are as follow:

| ES | 0.95 |
|----|------|
| DE | 0.83 |
| GT | 1.00 |
| PP1 | 1.00 |
| PP2 | 1.00 |

For downstream analysis, ATAC-seq peaks were merged from two independent differentiations for ES, DE, GT, PP1, and PP2 stage cells and from four donors for primary islets. Primary islet ATAC-seq data was obtained from previously published datasets (*Greenwald et al., 2019*).

## Hi-C data analysis

Hi-C data were processed as previously described with some modifications (*Dixon et al., 2015*). Read pairs were aligned to the hg19 reference genome separately using BWA-MEM with default parameters (*Li and Durbin, 2009*). Specifically, chimeric reads were processed to keep only the 5' position and reads with low mapping quality (<10) were filtered out. Read pairs were then paired using custom scripts. Picard tools were then used to remove PCR duplicates. Bam files with alignments were further processed into text format as required by Juicebox tools (*Durand et al., 2016*). Juicebox tools were then applied to generate hic files containing normalized contact matrices. All downstream analysis was based on 10 Kb resolution KR normalized matrices.

Chromatin loops were identified by comparing each pixel with its local background, as described previously (*Rao et al., 2014*) with some modifications. Specifically, only the donut region around the pixel was compared to model the expected count. Briefly, the KR-normalized contact matrices at 10

Kb resolution were used as input for loop calling. For each pixel, distance-corrected contact frequencies were calculated for each surrounding bin and the average of all surrounding bins. The expected counts were then transformed to raw counts by multiplying the counts with the raw-to-KR normalization factor. The probability of observing raw expected counts was calculated using Poisson distribution. All pixels with p-value<0.01 and distance less than 10 Kb were selected as candidate pixels. Candidate pixels were then filtered to remove pixels without any neighboring candidate pixels since they were likely false positives. Finally, pixels within 20 Kb of each other were collapsed and only the most significant pixel was selected. The collapsed pixels with p-value<1e-5 were used as the final list of chromatin loops.

A full set of scripts used for processing Hi-C data (*Qiu, 2021*) is available at https://github.com/MSanderlab/Pancreatic-progenitor-epigenome-maps-prioritize-type-2-diabetes-risk-genes-with-roles-in-development/tree/master (copy archived at swh:1:rev:ba79c687523c2696ea0ef30d8476e28a0d860f18).

## Definition of chromatin states

We collected or generated H3K4me1, H3K27ac, H3K4me1, H3K4me3, H3K27me3, and CTCF ChIP-seq data at each developmental stage and in mature islets. Data corresponding to mature islets was downloaded from previously published studies (*Bhandare et al., 2010*; *Parker et al., 2013*; *Pasquali et al., 2014*). Sequence reads were mapped to the human genome hg19 using bwa (version 0.7.12) (*Li and Durbin, 2009*), and low quality and duplicate reads were filtered using samtools (version 1.3) (*Li et al., 2009*). Using these reads, we then called chromatin states jointly across all data using chromHMM (version 1.12) (*Ernst and Kellis, 2012*) and used a 10-state model and 200 bp bin size, as models with larger state numbers did not empirically resolve any additional informative states. We then assigned state names based on patterns defined by the NIH Epigenome Roadmap (*Kundaje et al., 2015*), which included active promoter/TssA (high H3K4me3, high H3K27ac), flanking TSS/TssFlnk1 (high H3K4me3), flanking TSS/TssFlnk 2 (high H3K4me3, high H3K27ac, high H3K4me1), bivalent Tss/TssBiv (high H3K27me3, high H3K4me3), poised enhancer/EnhP (high H3K4me1), insulator/CTCF (high CTCF), active enhancer/EnhA (high H3K27ac, high H3K4me1), repressor (high H3K27me3), and two quiescent (low signal for all assays) states. The state map with assigned names is shown in *Figure 1—figure supplement 1A*.

We next defined stretch enhancer elements at each developmental stage and in mature islets. For each active enhancer (EnhA) element, we determined the number of consecutive 200 bp bins covered by the enhancer. We then modeled the resulting bin counts for enhancers in each cell type using a Poisson distribution. Enhancers with a p-value less than. 001 were labeled as stretch enhancers and otherwise labeled as traditional enhancers.

## Permutation-based significance

A random sampling approach (10,000 iterations) was used to obtain null distributions for enrichment analyses, in order to obtain p-values. Null distributions for enrichments were obtained by randomly shuffling enhancer regions using BEDTools (*Quinlan and Hall, 2010*) and overlapping with ATAC-seq peaks. p-values<0.05 were considered significant.

## Assignment of enhancer target genes

Transcriptomes were filtered for genes expressed (FPKM $\geq$1) at each relevant stage, and BEDTools (*Quinlan and Hall, 2010*) was used to assign each enhancer to the nearest annotated TSS.

## Gene ontology

All gene ontology analyses were performed using Metascape (*Zhou et al., 2019*) with default parameters.

## Motif enrichment analysis

The findMotifsGenome.pl. command in HOMER (*Heinz et al., 2010*) was used to identify enriched transcription factor binding motifs. de novo motifs were assigned to transcription factors based on suggestions generated by HOMER.

## T2D-relevant trait enrichment analysis

GWAS summary statistics for T2D (*Mahajan et al., 2018*; *Perry et al., 2012*), metabolic traits (HOMA-B, HOMA-IR [*Dupuis et al., 2010*], fasting glucose, fasting insulin [*Manning et al., 2012*], fasting proinsulin [*Strawbridge et al., 2011*], 2 hr glucose adjusted for BMI [*Saxena et al., 2010*], HbA1c, insulin secretion rate, disposition index, acute insulin response, peak insulin response [*Wood et al., 2017*]), and developmental traits (head circumference [*Taal et al., 2012*], birth length [*van der Valk et al., 2015*], birth weight [*Horikoshi et al., 2016*]) conducted with individuals of European ancestry were obtained from various sources including the MAGIC consortium, EGG consortium, and authors of the studies. Custom LD score annotation files were created for PSSE, PP2 stretch enhancers, and islet stretch enhancers using LD score regression version 1.0.1 (*Bulik-Sullivan et al., 2015*). Enrichments for GWAS trait-associated variants within PSSE, PP2 stretch enhancers, and islet stretch enhancers were estimated with stratified LD score regression (*Finucane et al., 2015*). We next determined enrichment in the proportion of variants in accessible chromatin sites within islet SE and PSSE with nominal association to beta cell-related glycemic traits. For each trait, we calculated a $2 \times 2$ table of variants mapping in and outside of islet SE or PSSE and with or without nominal association and then determined significance using a chi-square test.

## Adipocyte differentiation analysis

Chromatin states for human adipose stromal cell (hASC) differentiation stages (1-4) were obtained from a published study (*Varshney et al., 2017*). PSSE were intersected with hASC chromatin states using BEDTools intersect (version 2.26.0) (*Quinlan and Hall, 2010*) with default parameters.

## Identification of T2D risk loci intersecting PSSE

T2D GWAS summary statistics were obtained from the DIAMANTE consortium (*Mahajan et al., 2018*). Intersection of variants and PSSE was performed using BEDTools intersect (version 2.26.0) (*Quinlan and Hall, 2010*) with default parameters. The adjusted significance threshold was set at $p < 4.66 \times 10^{-6}$ (Bonferroni correction for 10,738 variants mapping in PSSE). Putative novel loci were defined as those with (1) at least one variant in a PSSE reaching the adjusted significance threshold and (2) mapping at least 500 kb away from a known T2D locus.

## ATAC-seq footprinting analysis

ATAC-seq footprinting was performed as previously described (*Aylward et al., 2018*). In brief, diploid genomes for CyT49 were created using vcf2diploid (version 0.2.6a) (*Rozowsky et al., 2011*) and genotypes called from whole genome sequencing and scanned for a compiled database of TF sequence motifs from JASPAR (*Mathelier et al., 2016*) and ENCODE (*ENCODE Project Consortium, 2012*) with FIMO (version 4.12.0) (*Grant et al., 2011*) using default parameters for p-value threshold and a 40.9% GC content based on the hg19 human reference genome. Footprints within ATAC-seq peaks were discovered with CENTIPEDE (version 1.2) (*Pique-Regi et al., 2011*) using cut-site matrices containing Tn5 integration counts within a ± 100 bp window around each motif occurrence. Footprints were defined as those with a posterior probability $\geq 0.99$.

## Generation of similarity matrices for total transcriptomes

For each replicate, FPKM values corresponding to total transcriptome were filtered for genes expressed (FPKM $\geq 1$) in $\geq 1$ replicate. For expressed genes, log(FPKM+1) values were used to calculate Pearson correlations.

## Immunofluorescence analysis

Cell aggregates derived from hESCs were allowed to settle in microcentrifuge tubes and washed twice with PBS before fixation with 4% paraformaldehyde (PFA) for 30 min at room temperature. Fixed samples were washed twice with PBS and incubated overnight at 4°C in 30% (w/v) sucrose in PBS. Samples were then loaded into disposable embedding molds (VWR), covered in Tissue-Tek O. C.T. Sakura Finetek compound (VWR) and flash frozen on dry ice to prepare frozen blocks. The blocks were sectioned at 10 µm and sections were placed on Superfrost Plus (Thermo Fisher) microscope slides and washed with PBS for 10 min. Slide-mounted cell sections were permeabilized and blocked with blocking buffer, consisting of 0.15% (v/v) Triton X-100 (Sigma) and 1% (v/v) normal

donkey serum (Jackson Immuno Research Laboratories) in PBS, for 1 hr at room temperature. Slides were then incubated overnight at 4℃ with primary antibody solutions. The following day slides were washed five times with PBS and incubated for 1 hr at room temperature with secondary antibody solutions. Cells were washed five times with PBS before coverslips were applied.

All antibodies were diluted in blocking buffer at the ratios indicated below. Primary antibodies used were goat anti-PDX1 (1:500 dilution, Abcam ab47383), mouse anti-NKX6.1 (1:300 dilution, Developmental Studies Hybridoma Bank F64A6B4), rabbit anti-Laminin (1:30, Sigma L-9393), and rabbit anti-CRB2 (1:500, Sigma SAB1301340). Secondary antibodies against goat and mouse were Alexa488- and Cy3-conjugated donkey antibodies, respectively (Jackson Immuno Research Laboratories 705-545-003 and 715-165-150, respectively), and were used at dilutions of 1:500 (anti-goat Alexa488) or 1:1000 (anti-mouse Cy3). Cell nuclei were stained with Hoechst 33342 (1:3000, Invitrogen). Representative images were obtained with a Zeiss Axio-Observer-Z1 microscope equipped with a Zeiss ApoTome and AxioCam digital camera. Figures were prepared in Adobe Creative Suite 5.

## Flow cytometry analysis

Cell aggregates derived from hESCs were allowed to settle in microcentrifuge tubes and washed with PBS. Cell aggregates were incubated with Accutase at room temperature until a single-cell suspension was obtained. Cells were washed with 1 mL ice-cold flow buffer comprised of 0.2% BSA in PBS and centrifuged at 200 g for 5 min. BD Cytofix/Cytoperm Plus Fixation/Permeabilization Solution Kit was used to fix and stain cells for flow cytometry according to the manufacturer's instructions. Briefly, cell pellets were re-suspended in ice-cold BD Fixation/Permeabilization solution (300 µL per microcentrifuge tube). Cells were incubated for 20 min at 4℃. Cells were washed twice with 1 mL ice-cold 1 × BD Perm/Wash Buffer and centrifuged at 10℃ and 200 × g for 5 min. Cells were re-suspended in 50 µL ice-cold 1 × BD Perm/Wash Buffer containing diluted antibodies, for each staining performed. Cells were incubated at 4℃ in the dark for 1–3 hr. Cells were washed with 1.25 mL ice-cold 1X BD Wash Buffer and centrifuged at 200 × g for 5 min. Cell pellets were re-suspended in 300 µL ice-cold flow buffer and analyzed in a FACSCanto II (BD Biosciences). Antibodies used were PE-conjugated anti-PDX1 (1:10 dilution, BD Biosciences); and AlexaFluor 647-conjugated anti-NKX6.1 (1:5 dilution, BD Biosciences). Data were processed using FlowJo software v10.

## Whole mount immunohistochemistry

Zebrafish larvae were fixed and stained according to published protocols (*Lancman et al., 2013*) using the following antibodies: chicken anti-GFP (1:200; Aves Labs; GFP-1020), guinea pig anti-insulin (1:200; Biomeda; v2024), mouse anti-Crb2a (1:100; ZIRC; zs-4), rabbit anti-panCrb (1:100; provided by Dr. Abbie M. Jensen at University of Massachusetts, Amherst; *Hsu and Jensen, 2010*), rabbit anti-Laminin (1:100; Sigma;L9393), mouse anti-Nkx6.1 (1:10; DSHB; F55A10), and DAPI (1:200; 500 mg/ml, Invitrogen; D1306).

## Imaging and quantification of beta cell numbers in zebrafish

To quantify beta cell numbers, 50 and 78 hpf zebrafish larvae were stained for confocal imaging using DAPI and guinea pig anti-insulin antibody (1:200; Biomeda; v2024). Whole mount fluorescent confocal Z-stacks (0.9 µm steps) images were collected for the entire islet with optical slices captured at a focal depth of 1.8 µm. Samples were imaged using a Zeiss 710 confocal microscope running Zen 2010 (Black) software. Final images were generated using Adobe Photoshop CS6 and/or ImageJ64 (vs.1.48b).

## Data sources

The following datasets used in this study were downloaded from the GEO and ArrayExpress repositories:

RNA-seq: Pancreatic differentiation of CyT49 hESC line (E-MTAB-1086); primary islet data (GSE115327).

ChIP-seq: H3K27ac data in primary islets (E-MTAB-1919 and GSE51311); H3K27ac data in pancreatic differentiation of CyT49 hESC line (GSE54471); H3K4me1 data in pancreatic differentiation of CyT49 hESC line (GSE54471); H3K4me1 data in primary islets (E-MTAB-1919 and E-MTAB 189);

H3K27me3 and H3K4me3 in pancreatic differentiation of CyT49 hESC line (E-MTAB-1086); H3K4me3 and H3K27me3 in primary islets (E-MTAB-189); CTCF in primary islets (E-MTAB-1919); PDX1 in CyT49 PP2 (GSE54471); samples from ROADMAP consortium: http://ncbi.nlm.nih.gov/geo/road-map/epigenomics.

ATAC-seq: primary islet data (PRJN527099); CyT49 PP2 (GSE115327).

Hi-C datasets were generated in collaboration with the Ren laboratory at University of California, San Diego as a component of the 4D Nucleome Project (*Dekker et al., 2017*) under accession number 4DNES0LVRKBM.

## Quantification and statistical analyses

Statistical analyses were performed using GraphPad Prism (v8.1.2), and R (v3.6.1). Statistical parameters, such as the value of n, mean, standard deviation (SD), standard error of the mean (SEM), significance level ($*p<0.05$, $**p<0.01$, and $***p<0.001$), and the statistical tests used, are reported in the figures and figure legends. The 'n' refers to the number of independent pancreatic differentiation experiments analyzed (biological replicates).

Statistically significant gene expression changes were determined with DESeq2 (*Love et al., 2014*).

## Acknowledgements

We thank Ileana Matta for assistance with ATAC-seq assays and library preparations, as well as the Sander and Gaulton laboratories for helpful discussions. We also thank Dr. Abbie Jensen at University of Massachusetts, Amherst for the anti-panCrb antibody. We acknowledge support of the UCSD Human Embryonic Stem Cell Core for cell sorting, as well as K Jepsen and the UCSD Institute for Genomic Medicine for library preparation and sequencing. This work was supported by NIH grants T32 GM008666 (RJG), P30 DK064391 (KJ, MS), R01 DK068471 (MS), 1DP2DK098092 (PDSD), and U01 DK105541 (MS, BR, PDSD); as well as the WM Keck Foundation 2017–01 (PDSD), and Diabetes Research Connection Project #08 (JJL)

## Additional information

### Competing interests

Kyle J Gaulton: This author consults for Genentech. The other authors declare that no competing interests exist.

### Funding

| Funder | Grant reference number | Author |
|---|---|---|
| National Institutes of Health | T32 GM008666 | Ryan J Geusz |
| National Institutes of Health | P30 DK064391 | Kyle J Gaulton<br>Maike Sander |
| National Institutes of Health | R01 DK068471 | Maike Sander |
| National Institutes of Health | U01 DK105541 | Bing Ren<br>P Duc Si Dong<br>Maike Sander |
| National Institutes of Health | 1DP2DK098092 | P Duc Si Dong |
| W.M. Keck Foundation | 2017-01 | P Duc Si Dong |
| Diabetes Research Connection | Project #08 | Joseph J Lancman |

The funders had no role in study design, data collection and interpretation, or the decision to submit the work for publication.

## Author contributions

Ryan J Geusz, Conceptualization, Data curation, Formal analysis, Validation, Investigation, Writing - original draft, Writing - review and editing; Allen Wang, Conceptualization, Data curation, Formal analysis, Validation, Investigation, Writing - review and editing; Joshua Chiou, Conceptualization, Formal analysis, Validation, Investigation, Writing - original draft, Writing - review and editing; Joseph J Lancman, Data curation, Formal analysis, Investigation, Writing - original draft, Writing - review and editing; Nichole Wetton, Jinzhao Wang, Jian Yan, Data curation; Samy Kefalopoulou, Data curation, Writing - review and editing; Yunjiang Qiu, Anthony Aylward, Formal analysis; Bing Ren, Supervision, Funding acquisition; P Duc Si Dong, Supervision, Writing - original draft, Writing - review and editing; Kyle J Gaulton, Conceptualization, Formal analysis, Supervision, Funding acquisition, Writing - original draft, Writing - review and editing; Maike Sander, Conceptualization, Resources, Supervision, Funding acquisition, Writing - original draft, Project administration, Writing - review and editing

## Author ORCIDs

Ryan J Geusz https://orcid.org/0000-0002-4897-9305
Allen Wang https://orcid.org/0000-0001-9870-7888
Joshua Chiou https://orcid.org/0000-0002-4618-0647
Joseph J Lancman https://orcid.org/0000-0003-1238-4446
Yunjiang Qiu https://orcid.org/0000-0002-0539-9714
Jian Yan http://orcid.org/0000-0002-1267-2870
Anthony Aylward http://orcid.org/0000-0003-3524-856X
Kyle J Gaulton https://orcid.org/0000-0003-1318-7161
Maike Sander https://orcid.org/0000-0001-5308-7785

## Ethics

Animal experimentation: Adult zebrafish and embryos were cared for and maintained under standard conditions. All research activity involving zebrafish was reviewed and approved by SBP Medical Discovery Institute Institutional Animal Care and Use Committee under protocol #18-014. hESC research was approved by the University of California, San Diego, Institutional Review Board and Embryonic Stem Cell Research oversight committee under project #090165ZX.

## Decision letter and Author response

Decision letter https://doi.org/10.7554/eLife.59067.sa1
Author response https://doi.org/10.7554/eLife.59067.sa2

# Additional files

## Supplementary files

• Transparent reporting form

## Data availability

All mRNA-seq, ChIP-seq, and ATAC-seq datasets generated for this study have been deposited at GEO under the accession number GSE149148. Source data files have been provided for Figures 2, 4, and 5.

The following dataset was generated:

| Author(s) | Year | Dataset title | Dataset URL | Database and Identifier |
|---|---|---|---|---|
| Geusz RJ, Wang A, Chiou J, Lancman JJ, Wetton N, Kefalopoulou S, Wang J, Qiu Y, Yan J, Aylward A, Ren | 2021 | Pancreatic progenitor epigenome maps prioritize type 2 diabetes risk genes with roles in development | https://www.ncbi.nlm.nih.gov/geo/query/acc.cgi?acc=GSE149148 | NCBI Gene Expression Omnibus, GSE149148 |

B, Si Dong PD,
Gaulton KJ, Sander
M

The following previously published datasets were used:

| Author(s) | Year | Dataset title | Dataset URL | Database and Identifier |
|---|---|---|---|---|
| Xie R, Everett LJ, Lim HW, Patel NA, Schug J, Kroon E, Kelly OG, Wang A, D'Amour KA, Robins AJ, Won KJ, Kaestner KH, Sander M | 2013 | ChIP-seq and RNA-seq of coding RNA of the progression of human embryonic stem cells to beta cells to characterize the epigenetic programs that underlie pancreas differentiation | https://www.ebi.ac.uk/arrayexpress/experiments/E-MTAB-1086/ | ArrayExpress, E-MTAB-1086 |
| Jin W, Mulas F, Gaertner B, Sui Y, Wang J, Matta I, Zeng C, Vinckier N, Wang A, Nguyen-Ngoc K, Chiou J, Kaestner KH, Frazer KA, Carrano AC, Shih H | 2019 | Identification of microRNA-dependent gene regulatory networks driving human pancreatic endocrine cell differentiation | https://www.ncbi.nlm.nih.gov/geo/query/acc.cgi?acc=GSE115327 | NCBI Gene Expression Omnibus, GSE115327 |
| Pasquali L, Gaulton KJ, Rodríguez-Seguí SA, Mularoni L, Miguel-Escalada I, Akerman I, Tena JJ, Morán I, Gómez-Marín C, van de Bunt M, Ponsa-Cobas J, Castro N, Nammo T, Cebola I, García-Hurtado J, Maestro MA, Pattou F, Piemonti L, Berney T, Gloyn AL, Ravassard P, Gómez-Skarmeta JL, Müller F, McCarthy MI, Ferrer J | 2014 | Pancreatic islet epigenomics reveals enhancer clusters that are enriched in Type 2 diabetes risk variants | https://www.ebi.ac.uk/arrayexpress/experiments/E-MTAB-1919/ | ArrayExpress, E-MTAB-1919 |
| Parker SC, Stitzel ML, Taylor DL, Orozco JM | 2013 | Chromatin stretch enhancer states drive cell-specific gene regulation and harbor human disease risk variants (ChIP-seq) | https://www.ncbi.nlm.nih.gov/geo/query/acc.cgi?acc=GSE51311 | NCBI Gene Expression Omnibus, GSE51311 |
| Wang A, Yue F, Li Y, Xie R | 2015 | Developmental Competence Encoded at the Level of Enhancers | https://www.ncbi.nlm.nih.gov/geo/query/acc.cgi?acc=GSE54471 | NCBI Gene Expression Omnibus, GSE54471 |
| Bhandare R, Schug J, Lay JL, Fox A, Smirnova O, Liu C, Naji A, Kaestner KH | 2010 | ChIP-Seq of human normal pancreatic islets with anti-histone antibodies to analyse histone modifications | https://www.ebi.ac.uk/arrayexpress/experiments/E-MTAB-189/ | ArrayExpress, E-MTAB-189 |
| University of California San Diego | 2015 | ATAC-seq in pancreatic islet cells | https://www.ncbi.nlm.nih.gov/bioproject/527099 | NCBI BioProject, PRJNA527099 |
| 4DN Network, Ren Laboratory | 2020 | Replicates of Hi-C on CyT49 cells | https://data.4dnucleome.org/experiment-set-replicates/4DNE- | 4D Nucleome, Sample4DNES0LVRKBM |

| | | | | |
|---|---|---|---|---|
| | | differentiated to pancreatic endoderm | SOLVRKBM/#processed-files | |
| Lister R, Pelizzola M, Dowen RH, Hawkins RD, Hon G, Tonti-Filippini J, Nery JR, Lee L, Ye Z, Ngo Q, Edsall L, Antosiewicz-Bourget J, Stewart R, Ruotti V, Millar AH, Thomson JA, Ren B, Ecker JR | 2009 | UCSD Human Reference Epigenome Mapping Projec | https://www.ncbi.nlm.nih.gov/geo/query/acc.cgi?acc=GSE16256 | NCBI Gene Expression Omnibus, GSE16256 |
| Bernstein BE, Stamatoyannopoulos JA, Costello JF, Ren B, Milosavljevic A, Meissner A, Kellis M, Marra MA, Beaudet AL, Ecker JR, Farnham PJ, Hirst M, Lander ES, Mikkelsen TJ, Thomson JA | 2010 | UCSF-UBC Human Reference Epigenome Mapping Project | https://www.ncbi.nlm.nih.gov/geo/query/acc.cgi?acc=GSE16368 | NCBI Gene Expression Omnibus, GSE16368 |
| Bernstein BE, Stamatoyannopoulos JA, Costello JF, Ren B, Milosavljevic A, Meissner A, Kellis M, Marra MA, Beaudet AL, Ecker JR, Farnham PJ, Hirst M, Lander ES, Mikkelsen TJ, Thomson JA | 2009 | BI Human Reference Epigenome Mapping Project | https://www.ncbi.nlm.nih.gov/geo/query/acc.cgi?acc=GSE17312 | NCBI Gene Expression Omnibus, GSE17312 |
| Bernstein BE, Stamatoyannopoulos JA, Costello JF, Ren B, Milosavljevic A, Meissner A, Kellis M, Marra MA, Beaudet AL, Ecker JR, Farnham PJ, Hirst M, Lander ES, Mikkelsen TJ, Thomson JA | 2010 | University of Washington Human Reference Epigenome Mapping Project | https://www.ncbi.nlm.nih.gov/geo/query/acc.cgi?acc=GSE18927 | NCBI Gene Expression Omnibus, GSE18927 |

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
