## [Decision Letter]

**Acceptance summary:**

The regard this as a well-designed study focused on developmental programs in pancreatic islet cells that underscore the risk of type 2 diabetes (T2D). Notably, it was found that progenitor-specific enhancers harboring T2D-associated variants regulate the cell polarity genes *LAMA1* and *CRB2*. Functionality was assessed by knocking down *lama1* or *crb2* in zebrafish embryos that led to defective pancreas morphogenesis and islet cell development. The studies unmask novel targets for therapeutic intervention for T2D.

**Decision letter after peer review:**

Thank you for submitting your article "Pancreatic progenitor epigenome maps prioritize type 2 diabetes risk genes with roles in development" for consideration by *eLife*. Your article has been reviewed by three peer reviewers, and the evaluation has been overseen by a Reviewing Editor and a Senior Editor. The reviewers have opted to remain anonymous.

The reviewers have discussed the reviews with one another and the Reviewing Editor has drafted this decision to help you prepare a revised submission.

Summary:

The risk of developing Type 2 diabetes (T2D) is influenced by a combination of genetic and environmental factors. For most genetic loci implicated in T2D risk, the disease-relevant target genes remain to be defined. In addition, it is not clear in which tissues the underlying genes exert their function. This manuscript explores through genetic analysis, the long-standing hypothesis that prior defect in beta cell differentiation may predispose to higher risk of developing T2D later in life. The authors used chromatin mapping of different developmental stages during directed development of beta cells from human ESCs. They convincingly identify stretch enhancers that are open and active at progenitor stages, but not in fully (or nearly fully) developed islets that contain T2D risk variants. In addition, gene-editing experiments demonstrate that 2 of the stretch enhancers are necessary for complete expression of their eponymous genes within the TADs of these enhancers, without any obvious other in vitro phenotype. Furthermore, when the eponymous genes, LMAT and CRB2, were knocked down in zebra fish, they were found to be critical for normal pancreatic morphogenesis and beta cell development. Overall the paper is well written, the data are convincing, and the conclusions are reasonable. A number of issues need to be addressed that require new experiments.

Essential revisions:

1) Since important stretch enhancers define cellular phenotype, it is not clear that expression of only genes within the enhancers' TAD are influenced by their activities (see for instance, Jian and Felsenfeld paper, PNAS, 2018: DOI: 10.1073/pnas.1803146115). Thus, it is felt critical that the authors perform RNAseq at the PP2 stage comparing control and the CRISPR-deleted stretch enhancer lines, to determine if there is influence on any other transcription factor(s) in development or beta cell function (or to see changes in in vitro phenotype). Further, since enhancers were deleted in the hESC experiment, and the genes were deleted in the zebra fish experiment, it would be helpful to include deletion of the genes in the ES context (since it is clearly difficult to find a syntenic enhancer in the fish model).

2) The authors modelled deletion of PSSEs associated with *LAMA1* and *CRB2* in an in vitro hESC differentiation system. Unfortunately, the mutations had no effect on the endocrine differentiation program in the hESC model. This may be due to the in vitro environment, where morphogenesis of the pancreas is not adequately recapitulated. To circumvent this problem, the authors used morpholino knockdowns in zebrafish embryos and found a reduced number of beta cells forming in the islet of the developing pancreas. Thus, they conclude that *LAMA1* and *CRB2* specifically affect pancreatic developmental programs, and suggest that aberrant developmental processes can predispose to T2D. In its present form, the work has potential but it requires essential additional data to support the central claims of the paper. Importantly, there are well-known issues with the anti-sense reagents, including unspecific effects and toxicity. Therefore, the consensus in the zebrafish field is that morpholino knockdown alone is an unreliable approach to assign new gene function. In fact, the number of beta cells that form in the zebrafish embryo would be highly sensitive to morpholino-related side-effects and toxicity. Therefore, the findings that *Lama1* and *Crb2b* play a role in beta-cell development is at best unsubstantiated. To make this claim conclusive, the authors would need to show similar phenotypes using homozygous mutants in *Lama1* and *Crb2b* or deletions in the corresponding enhancers in zebrafish. Once a "clean" phenotype is demonstrated, it will be important to address several critical questions to strengthen the current analysis: (a) Is the reduction in beta-cell number due to a defect in beta-cell proliferation, differentiation or both? (b) Does *Lama1* or *Crb2b* deficiency affect the specification, expansion or morphogenesis of the progenitor domains in the extra-pancreatic and/or intra-pancreatic ducts? (c) do the genes play a role in the formation of the dorsal- or ventral-bud-derived beta-cells or both? (d) Do the genes play a role in endocrine-cell migration, coalescence or clustering of the islet?

and (e) Does the reduction in the number of beta cells have a functional consequence for glucose tolerance in the fish, either under homeostasis or upon metabolic challenges, such as high glucose incubation or egg-yolk feeding?

3) The mammalian antibodies used for IHC (anti-*Lama1*, anti-pan-Crb, and anti-pan-*Crb2b*) have not been validated in fish. The IHC should be repeated in *Lama1* and *Crb2b* mutants or morphants. There should be a loss of signal in the pancreas if everything is as expected.

---

## [Author Response]

Essential revisions:1) Since important stretch enhancers define cellular phenotype, it is not clear that expression of only genes within the enhancers' TAD are influenced by their activities (see for instance, Jian and Felsenfeld paper, PNAS, 2018: DOI: 10.1073/pnas.1803146115). Thus, it is felt critical that the authors perform RNAseq at the PP2 stage comparing control and the CRISPR-deleted stretch enhancer lines, to determine if there is influence on any other transcription factor(s) in development or bet cell function (or to see changes in in vitro phenotype).

We apologize that we did not make it clear that we had already generated such datasets and performed the suggested analyses. RNA-seq data comparing control and CRISPR-deleted *LAMA1* stretch enhancer lines (Δ*LAMA1*Enh) at the PP2 stage are shown in Figure 4—figure supplement 1D, E. As shown in panel E we found minimal differences in the expression of transcription factors relevant to pancreas development between Δ*LAMA1*Enh and control PP2 cells, which led us to conclude that the enhancer deletion does not perturb pancreatic lineage induction. Similar RNA-seq analysis is shown comparing control and CRISPR-deleted *CRB2* stretch enhancer lines (Δ*CRB2*Enh) at the PP2 stage in Figure 5—figure supplement 2D, E. Combined, these analyses argue against a developmental phenotype resulting from the enhancer deletions, even when considering genes outside of each enhancer’s topologically associated domain. The list of all differentially expressed genes in *LAMA1* and *CRB2* enhancer-deleted PP2 cells is included as Figure 4—source data 1 and 2 and Figure 5—source data 1, respectively.

Further, since enhancers were deleted in the hESC experiment, and the genes were deleted in the zebra fish experiment, it would be helpful to include deletion of the genes in the ES context (since it is clearly difficult to find a syntenic enhancer in the fish model).

We agree that deletion of *LAMA1* and *CRB2* in hESCs would help better link the hESC system to the zebrafish results. Therefore, we generated and characterized clonal lines harboring deletions of the *LAMA1* (Δ*LAMA1*) and *CRB2* (Δ*CRB2*) genes, respectively.

In agreement with our observations in the Δ*LAMA1*Enh and Δ*CRB2*Enh lines, we found that deletion of *LAMA1* and *CRB2* does not perturb pancreatic lineage induction, as evidenced by the number of PDX1- and NKX6.1-positive cells and the expression of transcription factors relevant to pancreas development. The results are shown in revised Figure 4—figure supplement 2 and Figure 5—figure supplement 3, respectively. These new findings highlight the need for an in vivo model to fully characterize the role of *LAMA1* and *CRB2* in pancreatic development.

Regarding modeling of the enhancer deletion in zebrafish, results of a Basic Local Alignment Search Tool (BLAST) analysis (Altschul et al., 1990), as well as examination of each enhancer region using the Evolutionarily Conserved Regions (ECR) Browser (Ovcharenko et al., 2004) revealed that the *LAMA1* and *CRB2* stretch enhancers are indeed not conserved in this species.

2) The authors modelled deletion of PSSEs associated with LAMA1 and CRB2 in an in vitro hESC differentiation system. Unfortunately, the mutations had no effect on the endocrine differentiation program in the hESC model. This may be due to the in vitro environment, where morphogenesis of the pancreas is not adequately recapitulated. To circumvent this problem, the authors used morpholino knockdowns in zebrafish embryos and found a reduced number of beta cells forming in the islet of the developing pancreas. Thus, they conclude that LAMA1 and CRB2 specifically affect pancreatic developmental programs, and suggest that aberrant developmental processes can predispose to T2D. In its present form, the work has potential but it requires essential additional data to support the central claims of the paper. Importantly, there are well-known issues with the anti-sense reagents, including unspecific effects and toxicity. Therefore, the consensus in the zebrafish field is that morpholino knockdown alone is an unreliable approach to assign new gene function. In fact, the number of beta cells that form in the zebrafish embryo would be highly sensitive to morpholino-related side-effects and toxicity. Therefore, the findings that Lama1 and Crb2b play a role in beta-cell development is at best unsubstantiated. To make this claim conclusive, the authors would need to show similar phenotypes using homozygous mutants in Lama1 and Crb2b or deletions in the corresponding enhancers in zebrafish.Once a "clean" phenotype is demonstrated, it will be important to address several critical questions to strengthen the current analysis:a) Is the reduction in beta-cell number due to a defect in beta-cell proliferation, differentiation or both?b) Does Lama1 or Crb2b deficiency affect the specification, expansion or morphogenesis of the progenitor domains in the extra-pancreatic and/or intra-pancreatic ducts?c) do the genes play a role in the formation of the dorsal- or ventral-bud-derived beta-cells or both?d) Do the genes play a role in endocrine-cell migration, coalescence or clustering of the islet? ande) Does the reduction in the number of beta cells have a functional consequence for glucose tolerance in the fish, either under homeostasis or upon metabolic challenges, such as high glucose incubation or egg-yolk feeding?

While it is true that morpholino knockdowns can have non-specific effects and give rise to phenotypes that are inconsistent with genetic deletions, prior publications have extensively validated the morpholinos used in our study against *lama1* (Icha et al., 2016; Pollard et al., 2006; Randlett et al., 2011; Sidhaye and Norden, 2017; Yanakieva et al., 2019), *crb2a* (Omori and Malicki, 2006; Watanabe et al., 2010), and *crb2b* (Omori and Malicki, 2006). These studies validated efficacy for transcript and protein knockdown (Omori and Malicki, 2006; Randlett et al., 2011; Sidhaye and Norden, 2017) and demonstrated that the morphants phenocopy corresponding genetic mutants (Omori and Malicki, 2006). Of note, our controls for all morpholino experiments were embryos treated with non-targeting morpholinos rather than untreated embryos, therefore accounting for possible morpholino-based toxicity. We have provided additional information in the Materials and methods section of our manuscript to detail the extent to which these morpholinos have been previously validated.

While we agree that the suggestions for in-depth mechanistic studies in *lama1* and *crb2b* zebrafish mutants are interesting and important, we find them to be well beyond the scope of this manuscript and could themselves be the focus of an entirely separate project and publication. We maintain that the objective of our study is to define a progenitor-specific contribution to T2D risk by integrating genome-wide association data with chromatin maps from stem cell models of human pancreatic development. In this context, our analysis of *lama1* and *crb2* zebrafish morphants serves to illustrate that pancreatic endocrine cell development is sensitive to *lama1* and *crb2* transcript levels. Given that individually most type 2 diabetes-associated regulatory variants have subtle effects on transcript levels, a detailed understanding of phenotypes caused by complete *lama1* and *crb2* loss-of function through genetic deletion in zebrafish is likely of minor relevance to understanding variant effects and mechanisms of T2D risk. In fact, we are concerned about the likely possibility that a complete loss-of-function of these critical structural genes in animals would lead to more profound developmental defects that would mask the more subtle pancreatic phenotypes revealed by partial knockdown and therefore prove inconclusive. These considerations drove our decision to use morpholinos rather than genetic mutants for this study. The suggested experiments would require a significant time commitment to generate and analyze “clean” mutants (>1 year) and substantial financial resources to address questions which we consider to be interesting, but ultimately would not alter the conclusions of our manuscript as a whole.

3) The mammalian antibodies used for IHC (anti-Lama1, anti-pan-Crb, and anti-pan-Crb2b) have not been validated in fish. The IHC should be repeated in Lama1 and Crb2b mutants or morphants. There should be a loss of signal in the pancreas if everything is as expected.

The anti-laminin antibody used in this study has been used in 105 zebrafish references. Relevant to our work, it has been validated, using the same morpholino for *lama1* as used in our study (Randlett et al., 2011; Sidhaye and Norden, 2017). Likewise, the antipan-Crb antibody has also been previously validated (Hsu and Jensen, 2010).

Nevertheless, we have stained respective morphants with anti-laminin, anti-*Crb2a*, and anti-pan-Crb antibodies. We observed an absence of signal for laminin (revised Figure 6—figure supplement 2) and *Crb2a* (Figure 6—figure supplement 3) throughout development in embryos treated with *lama1-* and *crb2a*-specific morpholinos, respectively.

When staining embryos treated with morpholinos targeting *Crb2a* and *Crb2b* with the anti-pan-Crb antibody, we find that signal is lost in the ventral pancreas, where Nkx6.1+ pancreatic progenitor cells reside (Figure 6—figure supplement 3). Although we find residual signal in the dorsal pancreas, we note that the anti-pan-Crb antibody not only detects *Crb2* proteins but also *Crb1* and *Crb3*. Therefore, it is possible that other Crb proteins are expressed in the dorsal pancreas, which would not be affected by morpholinos targeting *Crb2a* and *Crb2b*.

In sum, our findings serve to validate both the efficacy of the morpholinos used in our knockdown experiments as well as the specificity of our selected antibodies.